# DIFFERENTIABLE RULE INDUCTION FROM RAW SEQUENCE INPUTS

**Kun Gao[1], Katsumi Inoue[2], Yongzhi Cao[3], Hanpin Wang[3], Feng Yang[1]***

[1]Institute of High Performance Computing, Agency for Science, Technology and Research
[2]National Institute of Informatics
[3]Key Laboratory of High Confidence Software Technologies, School of Computer Science
Peking University
`gaok@ihpc.a-star.edu.sg, inoue@nii.ac.jp,`
`{caoyz,whpxhy}@pku.edu.cn, yangf@ihpc.a-star.edu.sg`

## ABSTRACT

Rule learning-based models are widely used in highly interpretable scenarios due to their transparent structures. Inductive logic programming (ILP), a form of machine learning, induces rules from facts while maintaining interpretability. Differentiable ILP models enhance this process by leveraging neural networks to improve robustness and scalability. However, most differentiable ILP methods rely on symbolic datasets, facing challenges when learning directly from raw data. Specifically, they struggle with explicit label leakage: The inability to map continuous inputs to symbolic variables without explicit supervision of input feature labels. In this work, we address this issue by integrating a self-supervised differentiable clustering model with a novel differentiable ILP model, enabling rule learning from raw data without explicit label leakage. The learned rules effectively describe raw data through its features. We demonstrate that our method intuitively and precisely learns generalized rules from time series and image data.

## 1 INTRODUCTION

The deep learning models have obtained impressive performances on tabular classification, time series forecasting, image recognition, etc. While in highly trustworthy scenarios such as health care, finance, and policy-making process (Doshi-Velez & Kim, 2017), lacking explanations for decision-making prevents the applications of these complex deep learning models. However, the rule-learning models have interpretability intrinsically to explain the classification process. Inductive logic programming (ILP) is a form of logic-based machine learning that aims to learn logic programs for generalization and interpretability from training examples and background knowledge (Cropper et al., 2022). Traditional ILP methods design deterministic algorithms to induce rules from symbolic data to more generalized formal symbolic first-order languages (Quinlan, 1990; Blockeel & De Raedt, 1998). However, these symbolic ILP methods face robustness and scalability problems when learning from large-scale and ambiguous datasets (Evans et al., 2021; Hocquette et al., 2024). With the sake of robustness of neural networks, the neuro-symbolic ILP by combining neural networks and ILP methods can learn from noisy data (Evans & Grefenstette, 2018; Manhaeve et al., 2018; Gao et al., 2022a) and can be applied to large-scale datasets (Yang et al., 2017; Gao et al., 2024; Phua & Inoue, 2024). However, existing neuro-symbolic ILP methods are mainly learned from discrete symbolic data or fuzzy symbolic data that the likelihoods are generated from a pre-trained neural network module (Evans et al., 2021; Shindo et al., 2023). Learning logic programs from raw data is prevented because of the explicit label leakage problem, which is common in neuro-symbolic research (Topan et al., 2021): The leakage happens by introducing labels of ground objects for inducing rules (Evans & Grefenstette, 2018; Shindo et al., 2023). In fact, generating rules to describe objects in raw data without label information is necessary, especially when some objects are easily overlooked or lack labels yet are important for describing the data.

---

*Corresponding author.

In this study, we introduce *Neural Rule Learner* (NeurRL), a framework designed to learn logic programs directly from raw sequences, such as time series and flattened image data. Unlike prior approaches prone to explicit label leakage, where input feature labels are extracted using pre-trained supervised neural networks and then processed with differentiable ILP methods to induce rules trained with input labels, our method bypasses the need for supervised pre-trained networks to generate symbolic labels. Instead, we leverage a pre-trained clustering model in an unsupervised manner to discretize data into distinct features. Subsequently, a differentiable clustering module and a differentiable rule-learning module are jointly trained under supervision using raw input labels. This enables the discovery of rules that describe input classes based on feature distributions within the inputs. As a result, the model achieves efficient training in a fully differentiable pipeline while avoiding the explicit label leakage issue. The contributions of this study include: (1) We formally define the ILP learning task from raw inputs based on the interpretation transition setting of ILP. (2) We design a fully differentiable framework for learning symbolic rules from raw sequences, including a novel interpretable rule-learning module with multiple dense layers. (3) We validate the model's effectiveness and interpretability on various time series and image datasets.

## 2 RELATED WORK

Inductive logic programming (ILP), introduced by Muggleton & Feng (1990); Muggleton & De Raedt (1994), learns logic programs from symbolic positive and negative examples with background knowledge. Inoue et al. (2014) proposed learning from interpretation transitions, and Phua & Inoue (2021) applied ILP to Boolean networks. Manhaeve et al. (2018); Evans & Grefenstette (2018) adapted neural networks for differentiable and robust ILP, while Gao et al. (2022b) introduced a neural network-based ILP model for learning from interpretation transitions. This model was later extended for scalable learning from knowledge graphs (Gao et al., 2022a; 2024). Similarly, Liu et al. (2024) proposed a deep neural network for inducing mathematical functions. In our work, we present a novel neural network-based model for learning logic programs from raw numeric inputs.

In the raw input domain, Evans & Grefenstette (2018) proposed $\partial$ILP to learn rules from symbolic relational data, using a pre-trained neural network to map raw input data to symbolic labels. Unlike $\partial$ILP, which enforces a strong language bias by predefining logic templates and limiting the number of atoms, NeurRL uses only predicate types as language bias. Similarly, Evans et al. (2021) used pre-trained networks to map sensory data to disjunctive sequences, followed by binary neural networks to learn rules. Shindo et al. (2023) introduced $\alpha$ILP, leveraging object recognition models to convert images into symbolic atoms and employing top-$k$ searches to pre-generate clauses, with neural networks optimizing clause weights. Our approach avoids pre-trained large-scale neural networks for mapping raw inputs to symbolic representations. Instead, we propose a fully differentiable framework to learn rules from raw sequences. Additionally, unlike the memory-intensive rule candidate generation required by $\partial$ILP and $\alpha$ILP, NeurRL eliminates this step, enhancing scalability.

Adapting autoencoder and clustering methods in the neuro-symbolic domain shows promise. Sansone & Manhaeve (2023) applies conventional clustering on input embeddings for deductive logic programming tasks. Misino et al. (2022) and Zhan et al. (2022) use autoencoders and embeddings to calculate probabilities for predefined symbols to complete deductive logic programming and program synthesis tasks. In our approach, we use an autoencoder to learn representations for sub-areas of raw inputs, followed by a differentiable clustering method to assign ground atoms to similar patterns. The differentiable rule-learning module then searches for rule embeddings with these atoms in a bottom-up manner (Cropper & Dumancic, 2022). Similarly, DIFFNAPS (Walter et al., 2024) also uses an autoencoder to build hidden features and explain raw inputs. Additionally, BotCL (Wang et al., 2023) uses attention-based features to explain the ground truth class. However, the logical connections between the features used in DIFFNAPS and BotCL to describe the ground truth class are unclear. In contrast, rule-based explainable models like NeurRL use feature conjunctions to describe the ground truth class.

Azzolin et al. (2023) use a post-hoc rule-based explainable model to globally explain raw inputs from local explanations. In contrast, our model directly learns rules from raw inputs, and NeurRL's performance is unaffected by other explainable models. Das et al. (1998) used clustering to split sequence data into subsequences and symbolize them for rule discovery. Our model combines clustering and rule-learning in a fully differentiable framework to discover rules from sequence and image data,

with the rule-learning module providing gradient information to prevent cluster collapse (Sansone, 2023), where very different subsequences are assigned to the same clusters. He et al. (2018) viewed similar subsequences, called motifs, as potential rule bodies. Unlike their approach, we do not limit the number of body atoms in a rule. Wang et al. (2019) introduced SSSL, which uses shapelets as body atoms to learn rules and maximize information gain. Our model extends this by using raw data subsequences as rule body atoms and evaluating rule quality with precision and recall, a feature absent in SSSL.

## 3 PRELIMINARIES

### 3.1 LOGIC PROGRAMS AND INDUCTIVE LOGIC PROGRAMMING

A first-order language $\mathcal{L} = (D, F, C, V)$ (Lloyd, 1984) consists of predicates $D$, function symbols $F$, constants $C$, and variables $V$. A term is a constant, variable, or expression $f(t_1, \ldots, t_n)$ with $f$ as an $n$-ary function symbol. An atom is a formula $p(t_1, \ldots, t_n)$, where $p$ is an $n$-ary predicate symbol. A ground atom (fact) has no variables. A literal is an atom or its negation; positive literals are atoms, and negative literals are their negations. A clause is a finite disjunction of literals, and a rule (definite clause) is a clause with one positive literal, e.g., $\alpha_h \vee \neg\alpha_1 \vee \neg\alpha_2 \vee \cdots \vee \neg\alpha_n$. A rule $r$ is written as: $\alpha_h \leftarrow \alpha_1, \alpha_2, \ldots, \alpha_n$, where $\alpha_h$ is the head (head($r$)), and $\{\alpha_1, \alpha_2, \ldots, \alpha_n\}$ is the body (body($r$)), with each atom in the body called a body atom. A logic program $P$ is a set of rules. In first-order logic, a substitution is a finite set $\{v_1/t_1, v_2/t_2, \ldots, v_n/t_n\}$, where each $v_i$ is a variable, $t_i$ is a term distinct from $v_i$, and $v_1, v_2, \ldots, v_n$ are distinct (Lloyd, 1984). A ground substitution has all $t_i$ as ground terms. The ground instances of all rules in $P$ are denoted as ground($P$).

The Herbrand base $B_P$ of a logic program $P$ is the set of all ground atoms with predicate symbols from $P$, and an interpretation $I$ is a subset of $B_P$ containing the true ground atoms (Lloyd, 1984). Given $I$, the immediate consequence operator $T_P \colon 2^{B_P} \to 2^{B_P}$ for a definite logic program $P$ is defined as: $T_P(I) = \{\text{head}(r) \mid r \in \text{ground}(P), \text{body}(r) \subseteq I\}$ (Apt et al., 1988). A logic program $P$ with $m$ rules sharing the same head atom $\alpha_h$ is called a same-head logic program. A same-head logic program with $n$ possible body atoms can be represented as a matrix $\mathbf{M}_P \in [0, 1]^{m \times n}$. Each element $a_{kj}$ in $\mathbf{M}_P$ is defined as follows (Gao et al., 2022a): If the $k$-th rule is $\alpha_h \leftarrow \alpha_{j_1} \wedge \cdots \wedge \alpha_{j_p}$, then $a_{kj_i} = l_i$, where $l_i \in (0, 1)$ and $\sum_{s=1}^{p} l_s = 1$ ($1 \leq i \leq p$, $1 < p$, $1 \leq j_i \leq n$, $1 \leq k \leq m$). If the $k$-th rule is $\alpha_h \leftarrow \alpha_j$, then $a_{kj} = 1$. Otherwise, $a_{kj} = 0$. Each row of $\mathbf{M}_P$ represents a rule in $P$, and each column represents a body atom. An interpretation vector $\mathbf{v}_I \in \{0, 1\}^n$ corresponds to an interpretation $I$, where $\mathbf{v}_I[i] = 1$ if the $i$-th ground atom is true in $I$, and $\mathbf{v}_I[i] = 0$ otherwise. Given a logic program with $m$ rules and $n$ atoms, along with an interpretation vector $\mathbf{v}_I$, the function $D_P \colon \{0, 1\}^n \to \{0, 1\}^m$ (Gao et al., 2024) calculate the Boolean value for the head atom of each rule in $P$. It is defined as::

$$D_P(\mathbf{v}_I) = \theta(\mathbf{M}_P \mathbf{v}_I), \tag{1}$$

where the function $\theta$ is a threshold function: $\theta(x) = 1$ if $x \geq 1$, otherwise $\theta(x) = 0$. For a same-head logic program, the Boolean value of the head atom $v(\alpha_h)$ is computed as $v(\alpha_h) = \bigvee_{i=1}^{m} D_P(\mathbf{v}_I)[i]$. Additionally, Gao et al. (2024) replaces $\theta(x)$ with a differentiable threshold function and use fuzzy disjunction $\widetilde{\bigvee}_{i=1}^{m} \mathbf{x}[i] = 1 - \prod_{i=1}^{m} \mathbf{x}[i]$ to calculate the Boolean value of the head atom in a same-head logic program with neural networks.

Inductive logic programming (ILP) aims to induce logic programs from training examples and background knowledge (Muggleton et al., 2012). ILP learning settings include learning from entailments (Evans & Grefenstette, 2018), interpretations (De Raedt & Dehaspe, 1997), proofs (Passerini et al., 2006), and interpretation transitions (Inoue et al., 2014). In this paper, we focus on learning from interpretation transitions: Given a set $E \subseteq 2^{B_P} \times 2^{B_P}$ of interpretation pairs $(I, J)$, the goal is to learn a logic program $P$ such that $T_P(I) = J$ for all $(I, J) \in E$.

### 3.2 SEQUENCE DATA AND DIFFERENTIABLE CLUSTERING METHOD

A raw input consists of an instance $(\mathbf{x}, \mathbf{y})$, where $\mathbf{x} \in \mathbb{R}^{T_1 \times T_2 \times \cdots \times T_d}$ represents real-valued observations with $d$ variables, $T_i$ indicates the length for $i$-th variable, and $\mathbf{y} \in \{0, 1, \ldots, u-1\}$ is the class label, with $u$ classes. A sequence input is a type of raw input where $\mathbf{x} \in \mathbb{R}^{T_1}$ is an ordered sequence of real-valued observations (Wang et al., 2019). A subsequence $\mathbf{s}_i$ of length $l$ from sequence

$\mathbf{x} = (x_1, x_2, \ldots, x_T)$ is a contiguous sequence $(x_i, \ldots, x_{i+l-1})$ (Das et al., 1998). All possible subsequence with length $l$ include $\mathbf{s}_1, \ldots, \mathbf{s}_{T-l+1}$.

Clustering can be used to discover new categories (Rokach & Maimon, 2005). In the paper, we adapt the differentiable $k$-means method (Fard et al., 2020) to group raw data $\mathbf{x} \in \mathbf{X}$. First, an autoencoder $A$ generates embeddings $\mathbf{h}_\gamma$ for $\mathbf{x}$, where $\gamma$ represents the parameters. Then, we set $K$ clusters to discretize all raw data $\mathbf{x}$. The representation of the $k$-th cluster is $\mathbf{r}_k \in \mathbb{R}^p$, with $p$ as the dimension, and $\mathcal{R} = \{\mathbf{r}_1, \ldots, \mathbf{r}_K\}$ as the set of all cluster representations. For any vector $\mathbf{y} \in \mathbb{R}^p$, the function $c_f(\mathbf{y}; \mathcal{R})$ returns the closest representation based on a fully differentiable distance function $f$. Then, the differentiable $k$-means problems are defined as follows:

$$\min_{\mathcal{R}, \gamma} \sum_{\mathbf{x} \in \mathbf{X}} f(\mathbf{x}, A(\mathbf{x}; \gamma)) + \lambda \sum_{k=1}^{K} f(\mathbf{h}_\gamma(\mathbf{x}), \mathbf{r}_k) G_{k,f}(\mathbf{h}_\gamma(\mathbf{x}), \alpha; \mathcal{R}), \tag{2}$$

where the parameter $\lambda$ regulates the trade-off between seeking good representations for $\mathbf{x}$ and the representations that are useful for clustering. The weight function $G$ is a differentiable minimum function proposed by Jang et al. (2017):

$$G_{k,f}(\mathbf{h}_\gamma(\mathbf{x}), \alpha; \mathcal{R}) = \frac{e^{-\alpha f(\mathbf{h}_\gamma(\mathbf{x}), \mathbf{r}_k)}}{\sum_{k'=1}^{K} e^{-\alpha f(\mathbf{h}_\gamma(\mathbf{x}), \mathbf{r}_{k'})}}, \tag{3}$$

where $\alpha \in [0, +\infty)$. A larger value of $\alpha$ causes the approximate minimum function to behave more like a discrete minimum function. Conversely, a smaller $\alpha$ results in a smoother training process[1].

## 4 METHODS

### 4.1 PROBLEM STATEMENT AND FORMALIZATION

In this subsection, we define the learning problem from raw inputs using the interpretation transition setting of ILP and describe the language bias of rules for sequence data. We apply a neuro-symbolic description method from Marconato et al. (2023) to describe a raw input: (i) assume the label $\mathbf{y}$ depends entirely on the state of $K$ symbolic concepts $B = (a_1, a_2, \ldots, a_K)$, which capture high-level aspects of $\mathbf{x}$, (ii) the concepts $B$ depend intricately on the sub-symbolic input $\mathbf{x}'$ and are best extracted using deep learning, (iii) the relationship between $\mathbf{y}$ and $B$ can be specified by prior knowledge $\mathcal{B}$, requiring reasoning during the forward computing of deep learning models.

In this paper, we treat each subset of raw input as a constant and each concept as a ground atom. For example, in sequence data $\mathbf{x} \in \mathbf{X}$, if a concept increases from the point 0 to 10, the corresponding ground atom is $\texttt{increase}(\mathbf{x}[0 : 10])$. We define the *body symbolization function* $L_b$ to convert a continuous sequence $\mathbf{x}$ into discrete symbolic concepts, so that the set of all symbolic concepts $B$ across all raw inputs $\mathbf{X}$ holds $B = L_b(\mathbf{X})$. For binary class raw inputs, we use target atom $h_t$ to represents the target class, where $h_t$ being true means the instance belongs to the positive class. Then, a *head symbolization function* $L_h$ maps an binary input $\mathbf{x}$ to a set of atoms: if the class of $\mathbf{x}$ is the target class $t$, then $L_h(\mathbf{x}) = \{h_t\}$; otherwise, $L_h(\mathbf{x}) = \emptyset$. Hence, a logic program $P$ describing binary class raw inputs consists of rules with the same head atom $h_t$[2]. We define the Herbrand base $B_P$ of a logic program as the set of all symbolic concepts $B$ in all raw inputs and the head atom $h_t$. For a raw input $\mathbf{x}$, $L_b(\mathbf{x})$ represents a set of symbolic concepts $I \subseteq B_P$, which can be considered an interpretation. The task of inducing a logic program $P$ from raw inputs based on learning from interpretation transition is defined as follows:

**Definition 1 (Learning from Binary Raw Input)** *Given a set of raw binary label inputs $\mathbf{X}$, learn a same-head logic program $P$, where $T_P(L_b(\mathbf{x})) = L_h(\mathbf{x})$ holds for all raw inputs $\mathbf{x} \in \mathbf{X}$.*

In the paper, we aim to learn rules to describe the target class with the body consisting of multiple sequence features. Each feature corresponds to a subsequence of the sequence data. Besides, each feature includes the pattern information and region information of the subsequence. Based on the

---

[1]We set $\alpha$ to 1000 in the whole experiments.

[2]We can transfer multiple class raw inputs to the binary class by setting the class of interest as the target class and the other classes as the negative class.

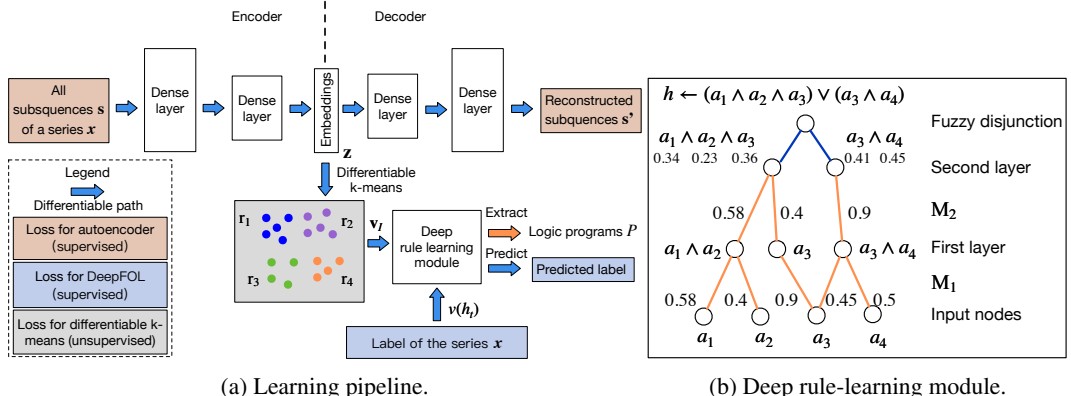

(a) Learning pipeline.  (b) Deep rule-learning module.

Figure 1: The learning pipeline of NeurRL and the rule-learning module.

pattern information, we further infer the mean value and tendency information of the subsequence. Using the region predicates, we can apply NeurRL to the dataset, where the temporal relationships between different patterns play a crucial role in distinguishing positive from negative examples. Specifically, we use the following rules to describe the sequence data with the target class:

$$h_t \leftarrow pattern_{i_1}(X_{j_1}), region_{k_1}(X_{j_1}), \ldots, pattern_{i_{n_1}}(X_{j_{n_2}}), region_{k_{n_3}}(X_{j_{n_2}}), \qquad (4)$$

where the predicate $\texttt{pattern}_i$ indicates the $i$-th pattern in all finite patterns within all sequence data, the predicate $\texttt{region}_k$ indicates the $k$-th region in all regions in a sequence, and the variable $X_j$ can be substituted by a subsequence of the sequence data. For example, $\texttt{pattern}_1(\mathbf{x}[0:5])$ and $\texttt{region}_0(\mathbf{x}[0:5])$ indicate that the subsequence $\mathbf{x}[0:5]$ matches the pattern with index one and belongs to the region with index zero, respectively. A pair of atoms, $\texttt{pattern}_i(X) \wedge \texttt{region}_k(X)$, corresponds to a feature within the sequence data. In this pair, the variables are identical, with one predicate representing a pattern and the other representing a region. We infer the following information from the rules in format (4): In a sequence input $\mathbf{x}$, if all pairs of ground patterns and regions atoms substituted by subsequences in $\mathbf{x}$ are true, then the sequence input $\mathbf{x}$ belongs to the target class represented by the head atom $h_t$.

## 4.2 DIFFERENTIABLE SYMBOLIZATION PROCESS

In this subsection, we design a differentiable body symbolization function $\tilde{L}_b$, inspired by differentiable $k$-means (Fard et al., 2020), to transform numeric sequence data $\mathbf{x}$ into a *fuzzy interpretation vector* $\mathbf{v}_I \in [0,1]^n$. This vector encodes fuzzy values of ground patterns and region atoms substituted by input subsequences. A higher value in $\mathbf{v}_I[i]$ indicates the $i$-th atom in the interpretation $I$ is likely true. Using the head symbolization function $J = L_h(\mathbf{x})$, we determine the target atom's Boolean value $v(h_t)$, where $v(h_t) = 1$ if $J = \{h_t\}$ and $v(h_t) = 0$ if $J = \emptyset$. Building on (Gao et al., 2024), we learn a logic program matrix $\mathbf{M}_P \in [0,1]^{m \times n}$ such that $\tilde{\bigvee}_{i=1}^m D_P(\mathbf{v}_I)[i] = v(h_t)$ holds. The rules in format (4) are then extracted from $\mathbf{M}_P$, generalized from the most specific clause with all pattern and region predicates.

The architecture of NeurRL is shown in Fig. 1a. To learn a logic program $P$ from sequence data, each input sequence $\mathbf{x}$ is divided into shorter subsequences $\mathbf{s}$ of length $l$ with a unit step stride. An encoder maps subsequences $\mathbf{s}$ to an embedding space $\mathbf{z}$, and a decoder reconstructs $\mathbf{s}'$ from $\mathbf{z}^3$. The differentiable $k$-means algorithm described in Section 3.2 clusters embeddings $\mathbf{z}$, grouping subsequences $\mathbf{s}$ with similar patterns into groups $\mathbf{r}$. This yields fuzzy interpretation vectors $\mathbf{v}_I$ and Boolean target atom value $v(h_t)$ for each sequence. Finally, the differentiable rule-learning module uses $\mathbf{v}_I$ as inputs and $v(h_t)$ as labels to learn high-level rules describing the target class.

Now, we describe the method to build the differentiable body symbolization function $\tilde{L}_b$ from sequence $\mathbf{x}$ to interpretation vector $\mathbf{v}_I$ as follows: Let $K$ be the maximum number of clusters based on the differentiable $k$-means algorithm, each subsequence $\mathbf{s}$ with the length $l$ in $\mathbf{x}$ and the corresponding embedding $\mathbf{z}$ has a cluster index $c$ $(1 \leq c \leq K)$, and each sequence input $\mathbf{x}$ can be transferred

---

³We use fully connected layers as the encoder and decoder structures.

into a vector of cluster indexes $\mathbf{c} \in \{0,1\}^{K \times (|\mathbf{x}|-l+1)}$. Additionally, to incorporate temporal or spatial information into the predicates of the target rules, we divide the entire sequence data into $R$ equal regions. The region of a subsequence $\mathbf{s}$ is determined by the location of its first point, $\mathbf{s}[0]$. For each subsequence $\mathbf{s}$, we calculate its *cluster index vector* $\mathbf{c}_s \in [0,1]^K$ using the weight function $G_{k,f}$ as defined Eq. (2), where $\mathbf{c}_s[k] = G_{k,f}(\mathbf{h}_\gamma(\mathbf{s}), \alpha; \mathcal{R})$ ($1 \le k \le K$ and $\sum_{i=1}^{K} \mathbf{c}_s[i] = 1$). A higher value in the $i$-th element of $\mathbf{c}_s$ indicates that the subsequence $\mathbf{s}$ is more likely to be grouped into the $i$-th cluster. Hence, we can transfer the sequence input $\mathbf{x} \in \mathbb{R}^T$ to *cluster index tensor* with the possibilities of cluster indexes of subsequences in all regions $\mathbf{c}_x \in [0,1]^{K \times l_p \times R}$, where $l_p$ is the number of subsequence in one region of input sequence data $\lceil |\mathbf{x}|/R \rceil$. To calculate the cluster index possibility of each region, we sum the likelihood of cluster index of all subsequence within one region in cluster index tensor $\mathbf{c}_x$ and apply softmax function to build *region cluster matrix* $\mathbf{c}_p \in \mathbb{R}^{K \times R}$ as follows:

$$\mathbf{c}[i,j] = \sum_{k=1}^{l_p} \mathbf{c}_x[i,k,j], \ \mathbf{c}_p[i,j] = \frac{e^{\mathbf{c}[i,j]}}{\sum_{i=1}^{K} e^{\mathbf{c}[i,j]}}. \tag{5}$$

Since the region cluster matrix $\mathbf{c}_p$ contains clusters index for each region. Besides, each cluster corresponds to a pattern. Hence, we flatten $\mathbf{c}_p$ into a fuzzy interpretation vector $\mathbf{v}_I$, which serves as the input to the deep rule-learning module of NeurRL.

## 4.3 Differentiable rule-learning module

In this section, we define the novel deep neural network-based rule-learning module denoted as $N_R$ to induce rules from fuzzy interpretation vectors. Based on the label of the sequence input $\mathbf{x}$, we can determine the Boolean value of target atom $v(h_t)$. Then, using fuzzy interpretation vectors $\mathbf{v}_I$ as inputs and Boolean values of the head atom $v(h_t)$ as labels $y$, we build a novel neural network-based rule-learning module as follows: Firstly, each input node receives the fuzzy value of a pattern or region atom stored in $\mathbf{v}_I$. Secondly, one output node in the final layer reflects the fuzzy values of the head atom. Then, the neural network consists of $k$ dense layers and one disjunction layer. Lastly, let the number of nodes in the $k$-th dense layer be $m$, then the forward computation process is formulated as follows:

$$\hat{y} = \overset{\sim}{\bigvee}_{i=1}^{m} \left( g_k \circ g_{k-1} \circ \cdots \circ g_1(\mathbf{v}_I) \right) [i], \tag{6}$$

where the $i$-th dense layer is defined as:

$$g_i(\mathbf{x}_{i-1}) = \frac{1}{1-d} \text{ReLU} \left( \mathbf{M}_i \mathbf{x}_{i-1} - d \right), \tag{7}$$

with $d$ as the fixed bias[4]. The matrix $\mathbf{M}_i$ is the softmax-activated of trainable weights $\tilde{\mathbf{M}}_i \in \mathbb{R}^{n_{out} \times n_{in}}$ in $i$-th dense layer of the rule-learning module:

$$\mathbf{M}_i[j,k] = \frac{e^{\tilde{\mathbf{M}}_i[j,k]}}{\sum_{u=1}^{n_{in}} e^{\tilde{\mathbf{M}}_i[j,u]}}. \tag{8}$$

With the differentiable body symbolization function $\tilde{L}_b$ and $N_R$, we now define a target function that integrates the autoencoder module, clustering module, and rule-learning module as follows:

$$\min_{\mathcal{R}, \gamma_e, \gamma_l} \sum_{\mathbf{s} \in \mathbf{x}, \mathbf{x} \in \mathbf{X}} f_1 \left( \mathbf{s}, A(\mathbf{s}; \gamma_e) \right) + \lambda_1 \sum_{k=1}^{K} f_1 \left( \mathbf{h}_{\gamma_e}(\mathbf{s}), \mathbf{r}_k \right) G_{k,f_1}(\mathbf{h}_{\gamma_e}(\mathbf{s}), \alpha; \mathcal{R}) + \lambda_2 f_2 \left( N_R \left( \tilde{L}_b(\mathbf{x}); \gamma_l \right), y \right), \tag{9}$$

where $\gamma_e$ and $\gamma_l$ represent the trainable parameters in the encoder and rule-learning module, respectively. The loss function $f_1$ and $f_2$ are set to mean square error loss and cross-entropy loss correspondingly. The parameters $\lambda_1$ and $\lambda_2$ regulate the trade-off between finding the concentrated representations of subsequences, the representations of clusters for obtaining precise patterns, and the representations of rules to generalize the data[5]. Fig. 1a illustrates the loss functions defined in

---

[4]In our experiments, we set the fixed bias as 0.5.

[5]In our experiments, we assigned equal weights to finding representations, identifying clusters, and discovering rules by setting $\lambda_1 = \lambda_2 = 1$.

the target function (9). The supervised loss functions are applied to the autoencoder (highlighted in orange boxes) and the rule-learning module (highlighted in blue boxes), respectively. The unsupervised loss function is applied to the differentiable k-means method (highlighted in gray box).

We analyze the interpretability of the rule-learning module $N_R$. In the logic program matrix $\mathbf{M}_P$ defined in Section 3.1, the sum of non-zero elements in each row, $\sum_{j=1}^{n} \mathbf{M}_P[i,j]$, is normalized to one, matching the threshold in the function $\theta$ in Eq. (1). The conjunction of the atoms corresponding to non-zero elements in each row of $\mathbf{M}_P$ can serve as the body of a rule. Similarly, in the rule-learning module of NeurRL, the sum of the softmax-activated weights in each layer is also one. Due to the properties of the activation function $\text{ReLU}(x-d)/(1-d)$ in each node, a node activates only when the sum of the weights approaches one; otherwise, it deactivates. This behavior mimics the threshold function $\theta$ in Eq. (1). Similar to the logic program matrix $\mathbf{M}_P$, the softmax-activated weights in each layer also have interpretability. When the fuzzy interpretation vector and Boolean value of the target atom fit the forward process in Eq. (6), the atoms corresponding to non-zero elements in each row of the $i$-th dense softmax-activated weight matrix $\mathbf{M}_i$ form a conjunction. From a neural network perspective, the $j$-th node in the $i$-th dense layer, denoted as $n_j^i$, represents a conjunction of atoms from the previous $(i-1)$-th dense layer (input layer). The likelihood of these atoms appearing in the conjunction is determined by the softmax-activated weights $\mathbf{M}_i[j,:]$ connecting to node $n_j^i$. In the final $k$-th dense layer, the disjunction layer computes the probability of the target atom $h_t$. The higher likelihood conjunctions, represented by the nodes in the final $k$-th layer, form the body of the rule headed by $h_t$. To interpret the rules headed by $h_t$, we compute the product of all softmax-activated weights $\mathbf{M}_i$ as the *program tensor*: $\mathbf{M}_P = \prod_{i=1}^{k} \mathbf{M}_i$, where $\mathbf{M}_P \in [0,1]^{m \times n}$. The program tensor has the same interpretability as the program matrix, with high-value atoms in each row forming the rule body and the target atom as the rule head. The number of nodes in the last dense layer $m$ determines the number of learned rules in one learning epoch. Fig. 1b shows a neural network with two dense layers and one disjunction layer in blue. The weights in orange represent significant softmax-activated values, with input nodes as atoms and hidden nodes as conjunctions. Multiplying the softmax weights identifies the atoms forming the body of a rule headed by the target atom.

We use the following method to extract rules from program tensor $\mathbf{M}_P$: We set multiple thresholds $\tau \in [0,1]$. When the value of the element in a row of $\mathbf{M}_P$ is larger than a threshold $\tau$, then we regard the atom corresponding to these elements as the body atom in a rule with the format (4). We compute the precision and recall of the rules based on the discretized fuzzy interpretation vectors generated from the test dataset. The discretized fuzzy interpretation vector is derived as the flattened version of $\bar{\mathbf{c}}_p[i,j] = \mathbb{1}(\mathbf{c}_p[i,j] = \max_k \mathbf{c}_p[k,j])$, where $\max_k \mathbf{c}_p[k,j]$ represents the maximum value in the $j$-th column of $\mathbf{c}_p$. Then, we keep the high-precision rules as the output. The precision and recall of a rule are defined as: $\text{precision} = n_{\text{body} \wedge \text{head}}/n_{\text{body}}$ and $\text{recall} = n_{\text{body} \wedge \text{head}}/n_{\text{head}}$, where $n_{\text{body} \wedge \text{head}}$ denotes the number of discretized fuzzy interpretation vectors $\mathbf{v}_I$ that satisfy the rule body and have the target class as the label. Similarly, $n_{\text{body}}$ represents the number of discretized fuzzy interpretation vectors that satisfy the rule body, while $n_{\text{head}}$ refers to the number of instances with the target class. When obtaining rules in the format (4), we can highlight the subsequences satisfying the pattern and region predicates above on the raw inputs for more intuitive interpretability.

To train the model, we first pre-train an autoencoder to obtain subsequence embeddings, then initialize the cluster embeddings using $k$-means clustering (Lloyd, 1982) based on these embeddings. Finally, we jointly train the autoencoder, clustering model, and rule-learning model using the target function (9) to optimize the embeddings, clusters, and rules simultaneously.

## 5 EXPERIMENTAL RESULTS

### 5.1 LEARNING FROM SYNTHETIC DATA

In this subsection, we evaluate the model on synthetic time series data based on triangular pulse signals and trigonometric signals. Each signal contains two key patterns, increasing and decreasing, with each pattern having a length of five units. To test NeurRL's learning capability on a smaller dataset, we set the number of inputs in both the positive and negative classes to two for both the training and test datasets. In each class, the difference between two inputs at each time point is a random number drawn from a normal distribution with a mean of zero and a variance of 0.1. The

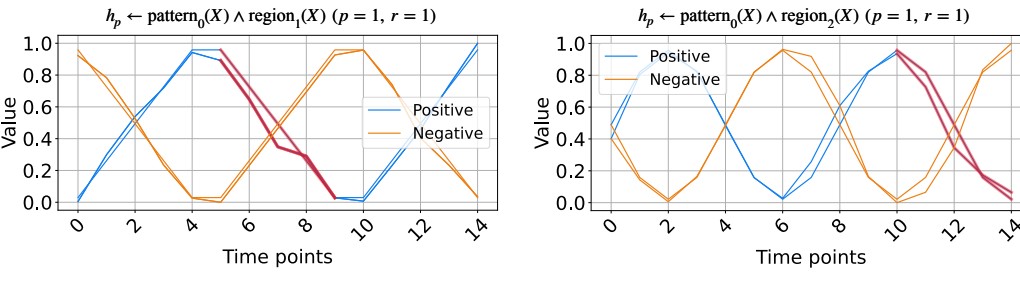

(a) The signals based on triangular pulse.  (b) The signals based on trigonometry function.

Figure 2: The synthetic data and the learned rules.

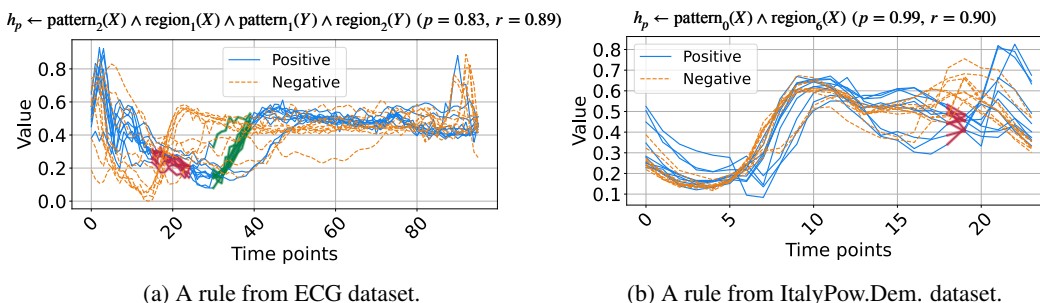

(a) A rule from ECG dataset.  (b) A rule from ItalyPow.Dem. dataset.

Figure 3: Selected rules from two UCR datasets.

positive test inputs are plotted in blue, and the negative test inputs in orange in Fig. 2. We set both the length of each region and the subsequence length to five units, and the number of clusters is set to three in this experiment. NeurRL is tasked with learning rules to describe the positive class, represented by the head atom $h_p$. If the ground body atoms, substituted by subsequences of an input, hold true, the head atom $h_p$ is also true, indicating that the target class of the input is positive.

The rules with 1.0 precision ($p$) and 1.0 recall ($r$) are shown in Fig. 2. The rule in Fig. 2a states that when $\texttt{pattern}_0$ (cluster index 0) appears in $\texttt{region}_1$ (from time points 5 to 9), the time series label is positive. Similarly, the rule in Fig. 2b indicates that when $\texttt{pattern}_0$ appears in $\texttt{region}_2$ (from time points 10 to 14), the label is positive. We highlight subsequences that satisfy the rule body in red in Fig. 2, inferring that the $\texttt{pattern}_0$ indicates decreasing. These red patterns perfectly distinguish positive from negative inputs.

## 5.2 LEARNING FROM UCR DATASETS

In this subsection, we experimentally demonstrate the effectiveness of NeurRL on 13 randomly selected datasets from UCR (Dau et al., 2019), as used by Wang et al. (2019). To evaluate NeurRL's performance, we consider the classification accuracy from the rules extracted by the deep rule-learning module (denoted as NeurRL(R)) and the classification accuracy from the module itself (denoted as NeurRL(N)). The number of clusters in this experiment is set to five. The subsequence length and the number of regions vary for each task. We set the number of regions to approximately 10 for time series data. Additionally, the subsequence length is set to range from two to five, depending on the specific subtask.

The baseline models include SSSL (Wang et al., 2019), Xu (Xu & Funaya, 2015), and BoW (Wang et al., 2013). SSSL uses regularized least squares, shapelet regularization, spectral analysis, and pseudo-labeling to auto-learn discriminative shapelets from time series data. Xu's method constructs a graph to derive underlying structures of time series data in a semi-supervised way. BoW generates a bag-of-words representation for time series and uses SVM for classification. Statistical details, such as the number of classes (C.), inputs (I.), series length, and comparison results are shown in

Table 1: Classification accuracy on 13 binary UCR datasets with different models.

| Dataset | C. | I. | Length | Xu | BoW | SSSL | NeurRL(R) | NeurRL(N) |
|---|---|---|---|---|---|---|---|---|
| Coffee | 2 | 56 | 286 | 0.588 | 0.620 | 0.792 | 0.964 | **1.000** |
| ECG | 2 | 200 | 96 | 0.819 | **0.955** | 0.793 | 0.820 | 0.880 |
| Gun point | 2 | 200 | 150 | 0.729 | **0.925** | 0.824 | 0.760 | 0.873 |
| ItalyPow.Dem. | 2 | 1096 | 24 | 0.772 | 0.813 | **0.941** | 0.926 | 0.923 |
| Lighting2 | 2 | 121 | 637 | 0.698 | 0.721 | **0.813** | 0.689 | 0.748 |
| CBF | 3 | 930 | 128 | 0.921 | 0.873 | **1.000** | 0.909 | 0.930 |
| Face four | 4 | 112 | 350 | 0.833 | 0.744 | 0.851 | 0.914 | **0.964** |
| Lighting7 | 7 | 143 | 319 | 0.511 | 0.677 | 0.796 | 0.737 | **0.878** |
| OSU leaf | 6 | 442 | 427 | 0.642 | 0.685 | 0.835 | 0.844 | **0.849** |
| Trace | 4 | 200 | 275 | 0.788 | **1.00** | **1.00** | 0.833 | 0.905 |
| WordsSyn | 25 | 905 | 270 | 0.639 | 0.795 | 0.875 | 0.932 | **0.946** |
| OliverOil | 4 | 60 | 570 | 0.639 | 0.766 | 0.776 | 0.768 | **0.866** |
| StarLightCurves | 3 | 9236 | 2014 | 0.755 | 0.851 | 0.872 | 0.869 | **0.907** |
| Mean accuracy | | | | 0.718 | 0.801 | 0.859 | 0.842 | **0.891** |

Table 2: Comparisons with non-differentiable $k$-means clustering algorithm.

| Dataset | Non-differentiable $k$-means | | Differentiable $k$-means | |
|---|---|---|---|---|
| | accuracy | running time (s) | accuracy | running time (s) |
| Coffee | 0.893 | 313 | **0.964** | **42** |
| ECG | 0.810 | 224 | **0.820** | **65** |
| Gun point | **0.807** | 102 | 0.740 | **35** |
| ItalyPow.Dem. | 0.845 | 114 | **0.926** | **63** |
| Lighting2 | 0.672 | 1166 | **0.689** | **120** |

Tab. 1, with the best results in bold and second-best underlined. The NeurRL(N) achieves the most best results, with seven, and NeurRL(R) achieves five second-best results.

The learned rules from the ECG and ItalyPow.Dem. datasets in the UCR archive are shown in Fig. 3. In Fig. 3a, red highlights subsequences with the shape $\texttt{pattern}_2$ in the region $\texttt{region}_1$, while green highlights subsequences with the shape $\texttt{pattern}_1$ in the region $\texttt{region}_2$. The rule suggests that when data decreases between time points 15 to 25 and then increases between time points 30 to 40, the input likely belongs to the positive class. The precision and recall for this rule are 0.83 and 0.89, respectively. In Fig. 3b, red highlights subsequences with the shape $\texttt{pattern}_0$ in the region $\texttt{region}_6$. The rule indicates that a lower value around time points 18 to 19 suggests the input belongs to the positive class, with precision and recall of 0.99 and 0.90, respectively.

To demonstrate the benefits of the fully differentiable learning pipeline from raw sequence inputs to symbolic rules, we compare the accuracy and running times (in seconds) between NeurRL and its deep rule-learning module using the non-differentiable $k$-means clustering algorithm (Lloyd, 1982). We use the same hyperparameter to split time series into subsequences for two methods. Results in Tab. 2 show that the differentiable pipeline significantly reduces running time without sacrificing rule accuracy in most cases.

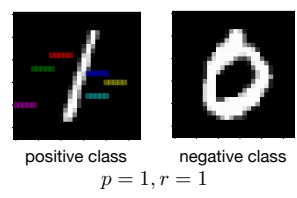

positive class     negative class

$p = 1, r = 1$

Figure 4: Learned rules from MNIST datasets.

### 5.3 LEARNING FROM IMAGES

In this subsection, we ask the model to learn rules to describe and discriminate two classes of images from MNIST datasets. We divide the MNIST dataset into five independent datasets, where each dataset contains one positive class and one negative class. For two-dimensional image data, we first flatten the image data to one-dimensional sequence data. Then, the

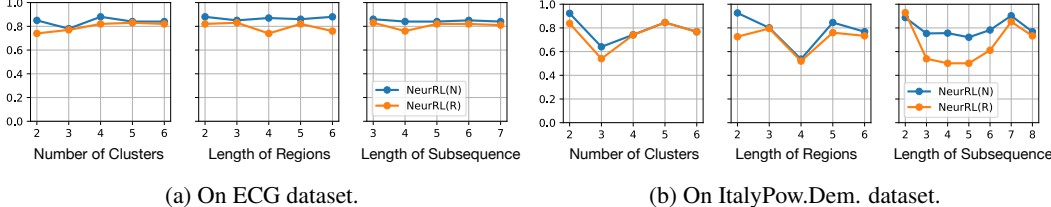

(a) On ECG dataset.  (b) On ItalyPow.Dem. dataset.

Figure 5: Results of ablation study. Hyperparameter values vs. accuracy.

sequence data can be the input for the NeurRL model to learn the rules. The lengths of subsequence and region are both set to three. Besides, the number of clusters is set to five. After generating the highlighted patterns based on the rules, we recover the sequence data to the image for interpreting these learned rules.

We present the rule for learning the digit one from the digit zero in Fig. 4, and the rules for learning other positive class from the negative class are shown in Fig. 6 in Appendix C. We present the rule with the precision larger than 0.9 and the highlight features (or areas) defined in the rules. Each highlight feature corresponds to a pair of region and pattern atoms in a learned rule. We can interpret rules like importance attention (Zhang et al., 2019), where the colored areas include highly discriminative information for describing positive inputs compared with negative inputs. For example, in Fig. 4, if the highlighted areas are in black at the same time, then the image class is one. Otherwise, the image class is zero. Compared with attention, we calculate the precision and recall to evaluate these highlight features quantitatively.

## 5.4 ABLATION STUDY

We conducted ablation studies using default hyperparameters, except for the one being explored. In this study, the number of clusters and regions influences the number of atoms in the learned rules, while the length of subsequences depends on the length of potential patterns. Hence, these three hyperparameters collectively describe the sensitivity of NeurRL. We present the accuracy of both NeurRL(N) and NeurRL(R) on the time series tasks ECG and ItalyPow.Dem. from the UCR archive.

From Fig. 5, we observe that NeurRL's sensitivity varies across tasks, and the subsequence length is a sensitive hyperparameter when the sequence length is small, as seen in the ItalyPow.Dem. dataset. Properly chosen hyperparameters can achieve high consistency in accuracy between rules and neural networks. Notably, increasing the number of clusters or length of regions does not reduce accuracy linearly. This suggests that the rule-learning module effectively adjusts clusters within the model. In addition, we conduct another ablation study in Tab. 3 of Appendix A to show that pre-training the autoencoder and clustering prevents cluster collapse (Sansone, 2023) and improves accuracy without increasing significant training time.

## 6 CONCLUSION

Inductive logic programming (ILP) is a rule-based machine learning method that supporting data interpretability. Differentiable ILP offers advantages in scalability and robustness. However, label leakage remains a challenge when learning rules from raw data, as neuro-symbolic models require intermediate feature labels as input. In this paper, we propose a novel fully differentiable ILP model, Neural Rule Learner (NeurRL), which learns symbolic rules from raw sequences using a differentiable $k$-means clustering module and a deep neural network-based rule-learning module. The differentiable $k$-means clustering algorithm groups subcomponents of inputs based on the similarity of their embeddings, and the learned clusters are used as input for the rule-learning module to induce rules that describe the ground truth class of input based on its features. Compared to other rule-based models, NeurRL achieves comparable classification accuracy while offering interpretability through quantitative metrics. Future goals include variables representing entire inputs to explain handwritten digits without label leakage (Evans et al., 2021). Another focus is learning from incomplete data (e.g., healthcare applications) to address real-world challenges.

ACKNOWLEDGMENTS

We express our gratitude to the anonymous Reviewers for their valuable feedback, which has contributed significantly to enhancing the clarity and presentation of our paper. We also thank Xingyao Wang, Yingzhi Xia, Yang Liu, Jasmine Ong, Justina Ma, Jonathan Tan, Yong Liu, and Rick Goh for their supporting.

This work has been supported by AI Singapore under Grant AISG2TC2022006, Singapore. This work has been supported by the NII international internship program, JSPS KAKENHI Grant Number JP21H04905 and JST CREST Grant Number JPMJCR22D3, Japan. This work has also been supported by the National Key R&D Program of China under Grant 2021YFF1201102 and the National Natural Science Foundation of China under Grants 61972005 and 62172016.

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

## A ABLATION STUDY ON PRE-TRAINING

In this section, we conduct experiments with and without pre-training the autoencoder and clustering model to investigate whether pre-training improves accuracy. The results, including accuracy and running time (in seconds), are presented in Tab. 3.

Table 3: Ablation study: With vs. without pre-training autoencoder and clustering model. The first accuracy is achieved using NeurRL(N), while the second accuracy is obtained with NeurRL(R).

| Dataset | With pre-training | | Without pre-training | |
| --- | --- | --- | --- | --- |
| | Accuracy | Running time (s) | Accuracy | Running time (s) |
| Coffee | **1.00**, **0.96** | 42 | 0.83, 0.81 | **30** |
| ECG | **0.88**, **0.82** | 65 | 0.87, 0.64 | **53** |
| ItalyPow.Dem. | **0.92**, **0.93** | 63 | 0.75, 0.80 | **61** |
| Gun Point | **0.87**, **0.76** | 35 | 0.86, 0.43 | **31** |
| Lighting2 | **0.75**, **0.69** | 120 | 0.64, 0.60 | **63** |

The experiment results indicate the pre-training autoencoder and clustering model can improve the accuracy of both NeurRL(N) and NeurRL(R) without increasing significant training time.

## B THE LINK OF THE MODEL

The model and data can be found here: `https://github.com/gaokun12/NeurRL`

## C CLASSIFYING AND EXPLAINING MNIST DATA

Each column in Fig. 6 corresponds to a learning task. There are six rules presented in Fig. 6 to describe the positive class from the negative class.

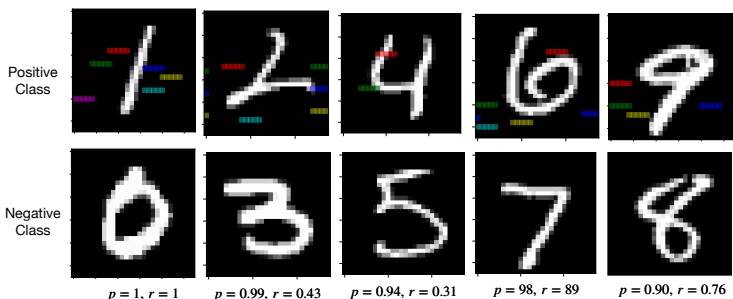

Figure 6: Rules from MNIST datasets.

