# OpenReview forum: "Differentiable Rule Induction from Raw Sequence Inputs"
_ICLR.cc/2025/Conference — ICLR 2025 Poster_

### Official Review · Reviewer_iHmD · 2024-10-24

**Soundness:** 3
**Presentation:** 3
**Contribution:** 3
**Rating:** 6
**Confidence:** 3

**Summary:**

This paper proposes a novel differentiable Inductive Logic Programming model called NeurRL, aimed at solving the problem of learning symbolic rules from raw data. Unlike previous methods that rely on pre-trained neural networks to extract labels for symbolic learning, NeurRL combines a differentiable clustering module and a deep neural network learning module, enabling end-to-end rule learning from raw data such as time series and images without leaking label information.

**Strengths:**

1. The paper formally establishes the ILP learning task from raw data, utilizing the interpretation transition framework of ILP to guide the learning process.
2. Experimental results demonstrate that NeurRL shows a notable improvement over previous methods in terms of effectiveness.
3. The paper's overall presentation is clear, making the concepts and methodologies easy to follow.

**Weaknesses:**

1. The model has a large number of hyperparameters (especially the number of categories K, subsequence length l, and the number of regions P), which may negatively affect reproducibility and generalizability.
2. The experimental details are insufficient, lacking information on how hyperparameters were chosen and whether tuning was done separately for different experiments. Moreover, it is unclear if the results reported are the best performance or averaged.
3. The learned predicates lack high-level abstraction, instead offering post-hoc explanations for neighboring pixels. This limits the generalizability and interpretability of the induced rules.
4. Although previous works did not formally define the task of extracting rules from raw data, some have achieved similar goals with comparable architectures (e.g., autoencoder + discriminator)[1, 2], or going further in technical sophistication[3] . This paper lacks a discussion of and comparison with such works.
5. The model does not consider translation invariance in the data, meaning some patterns should only need to be present rather than bound to a specific region. This limitation may hinder the model's performance in tasks where the position of the target is irrelevant.

[1] Azzolin, S., Longa, A., Barbiero, P., Liò, P., & Passerini, A. (2022). Global explainability of gnns via logic combination of learned concepts. arXiv preprint arXiv:2210.07147.

[2] Walter, N. P., Fischer, J., & Vreeken, J. (2024, March). Finding interpretable class-specific patterns through efficient neural search. In Proceedings of the AAAI Conference on Artificial Intelligence (Vol. 38, No. 8, pp. 9062-9070).

[3] Wang, B., Li, L., Nakashima, Y., & Nagahara, H. (2023). Learning bottleneck concepts in image classification. In Proceedings of the ieee/cvf conference on computer vision and pattern recognition (pp. 10962-10971).

**Questions:**

1. Is there a way to reduce the number of hyperparameters, such as determining the number of categories K adaptively during training? If this is not feasible, please at least provide guidance on how to select these hyperparameters and show how different choices impact the experimental results.
2. Please compare your work with more related studies (see Weakness 4)

---

> ### Author Response · Authors · 2024-11-22
> **Responses to the Weaknesses Part 1**
>
> Thank you for your insightful review. We have addressed each of your comments with detailed responses provided after each one.
>
> >Weaknesses:
> >The model has a large number of hyperparameters (especially the number of categories K, subsequence length l, and the number of regions P), which may negatively affect reproducibility and generalizability.
> >The experimental details are insufficient, lacking information on how hyperparameters were chosen and whether tuning was done separately for different experiments. Moreover, it is unclear if the results reported are the best performance or averaged.
>
> We appreciate your proposed advice. We appended several statements in line 392, lines 420 to 428, and line 501 of Section 5 to explain and specify what exactly the hyperparameter we used in the experiments.
>
> Specifically, there are three essential hyperparameters, including the number of clusters, the length of subsequence, and the length of the period.  Please note that the length of a period is subject to the number of periods because of the fixed length of series data. The number of clusters indicates the number of atoms in the hypothesis. The length of the subsequence is related to the length of the potential pattern in the raw inputs. Hence, we use separate hyperparameters in different tasks in UCR tasks and MNIST data. For the number of clusters, we set it as three, five, and five in synthetic data, UCR archive, and image datasets, respectively.  For the length of the subsequence, we set it to five in synthetic data. In UCR data, the length of subsequence ranges from two to five in different tasks. In MNIST data, we set the length of the subsequence to three. For the length of the period, we set it to five in the synthetic dataset. In the UCR dataset, because the time series length for different tasks is different, we roughly constrain the number of the period to 10. In the MNIST dataset, the length of the single period is set to three.
>
> In conclusion, the choosing hype parameter is related to the explainability granularity based on each sequence data with different lengths. We present the optimal results in Table 1.
>
>
> > The learned predicates lack high-level abstraction, instead offering post-hoc explanations for neighboring pixels. This limits the generalizability and interpretability of the induced rules.
>
> Thank you for proposing these comments. In section 5, we intuitively explain the semantics of the generated rules by plotting the rules in the corresponding raw data.
> Besides this, we also illustrate the semantics of a rule for describing the raw data in Section 4.1. We highlighted that we can use representative features as atoms in rules to describe the ground truth class. In addition, the temporal information of the representative patterns should also be important to generalize the knowledge. With the period predicates, the rules include the temporal relationships between different representative patterns. Hence, we believe the rules with the well-defined language bias can describe any raw data.
>
> Besides, as the most challenging task in artificial intelligence, creating unseen symbols to automatically generalize these representative patterns in a rule can make the semantics of the rule clearer. To solve this problem, we may need natural language processing techniques, and we will keep this problem as a future task by adopting state-of-the-art large language models.

---

> ### Author Response · Authors · 2024-11-22
> **Responses to the Weaknesses Part 2**
>
> > Although previous works did not formally define the task of extracting rules from raw data, some have achieved similar goals with comparable architectures (e.g., autoencoder + discriminator)[1, 2], or going further in technical sophistication[3] . This paper lacks a discussion of and comparison with such works.
> >
> > [1] Azzolin, S., Longa, A., Barbiero, P., Liò, P., & Passerini, A. (2022). Global explainability of gnns via logic combination of learned concepts. arXiv preprint arXiv:2210.07147.
> >
> > [2] Walter, N. P., Fischer, J., & Vreeken, J. (2024, March). Finding interpretable class-specific patterns through efficient neural search. In Proceedings of the AAAI Conference on Artificial Intelligence (Vol. 38, No. 8, pp. 9062-9070).
> >
> > [3] Wang, B., Li, L., Nakashima, Y., & Nagahara, H. (2023). Learning bottleneck concepts in image classification. In Proceedings of the ieee/cvf conference on computer vision and pattern recognition (pp. 10962-10971).
>
> Thank you for supporting the related reference. We have added these references to our discussion work. Specifically, Azzolin et al. [1] propose GLGExplainer to learn global explainable rules based on the local explanations. Although both models produce rules as global explanation results, the rule-learning methods, model pipelines, and input data differ significantly. Firstly, NeurRL includes a novel fully differentiable rule learner based on the propositionalization method, which is a solution in inductive logic programming research. GLGExplainer uses a post-hoc rule-based explanation method [4] to explain the prediction behavior of neural networks. Hence, our rule learning method is data-driven and does not rely on the performance of well-trained neural networks compared with post-hoc explanation methods. Secondly, our model learns global explanations from raw data directly, which is a data-driven rule-learning model. However, GLGExplainer learns rules from local explanations generated from another model. Hence, our model's performance also won't rely on the performance of local explanation models.  Thirdly, our model NeurRL learns global explanations in rules to explain ground truth class with the possible features inside the raw time series data or image data. However, GLGExplainer explains graph data based on graph neural networks.
>
> Compared with DIFFNAPS proposed by Walter et al. [2], even though our work NeurRL and DIFFNAPS both use autoencoder to find patterns represented by the hidden embeddings, the interpretability granularity is different. In our explainability, we present the patterns stipulated in rules in the original raw data and illustrate patterns with region information. In addition, we can describe the ground truth class with conjunction of patterns with region information. Hence, we can easily evaluate the learned rules with precision and recall metrics. On the contrary, the DIFFNAPS can only output the important patterns that are related to the ground truth class. DIFFNAPS cannot explicitly output information about what kind of logic combinations of patterns can describe the ground truth class. Hence, verifying the soundness of interpretability results needs a lot of domain knowledge in real tasks.
>
> Compared with BotCL [3] proposed by Wang et al., which uses attention mechanisms, feature aggregation, classification, and reconstruction loss to provide explanations for images. Compared with the attention model-based explainable AI models, our rule learning model is a white-box model.  The forward computing process in the rule learning module in NeurRL imitates the logic programming process. Hence, the learned rules can be evaluated quantitatively with precision and recall values. Furthermore, our explainable results, expressed as rules, also capture the information of pattern conjunctions that describe the ground truth class. However, the BotCL only highlights single patterns through attention to describe an instance with its ground truth class.

---

> ### Author Response · Authors · 2024-11-22
> **Responses to the Weaknesses Part 3, Questions, and Reference**
>
> > The model does not consider translation invariance in the data, meaning some patterns should only need to be present rather than bound to a specific region. This limitation may hinder the model's performance in tasks where the position of the target is irrelevant.
>
> Thank you for considering these problems. Our model can learn rules with patterns which is time-independent. We consider region information for the following reasons: Firstly, when a pattern occurring in a specific time can only describe the ground truth class, then the rules with region information are very necessary. Secondly, when using region predicate, we can also describe the temporal relations such as 'before' between the patterns in different regions.
>
> Besides, there are two alternative methods to learn time-invariant patterns. Firstly, for the patterns that are time-independent, our rule learner may learn multiple rules that include the same patterns but many different region predicates. Hence, with the least generalization algorithm [5], we can easily induce a more generalized rule with time-invariant patterns. Secondly, the input nodes of the rule learning neural networks correspond to the pattern predicates and period predicates, which can be found in Eq. (6) and the region cluster matrix $\mathbf{c}_p$ defined in Section 4.2. Then, the rule-learning neural networks are easy to modify to learn rules with only pattern predicates. Specifically, we replace the flattened vector from the region cluster matrix $\mathbf{c}_p$ with the *global pattern vector* $\mathbf{c}_n$ as the input for the rule-learning module. The global pattern vector $\mathbf{c}_n$ can be calculated by the following operation: $\mathbf{c}_n[i] = \sum _{x=1}^P \mathbf{c}_p[i,x] (1\leq i \leq K)$, where $P$ is the number of periods, and $K$ is the number of clusters.
>
> > Questions:
> > Is there a way to reduce the number of hyperparameters, such as determining the number of categories K adaptively during training? If this is not feasible, please at least provide guidance on how to select these hyperparameters and show how different choices impact the experimental results.
>
> Thank you for these suggestions. Three types of hyperparameters need to be tuned in NeurRL, including The number of clusters, the length of subsequence, and the length of the period.  The length of the period affects the number of input nodes in the rule-learning module. Hence, the length of the period may affect the performance of the rule-learning module. Because the length of the period is very related to the length of the sequence, we chose the number of periods to be roughly 10 throughout all the UCR tasks. In the MNIST dataset, we set the length of the period to be equal to the length of the subsequence. When the length of the period is equal to the length of the subsequent, we will omit the temporal relations of different patterns inside the same period. In addition, we set the number of periods to five in the synthetic dataset, which is equal to the length of each pattern in the synthetic data. The number of clusters can be set to 5, which is not very sensitive to affect the model's performance because the rule learning model can collapse some clusters if the number of clusters is too large. For the length of the subsequence, we set it very small, from 3 to 5, in all tasks for the clustering model to capture the pattern easily. We will consider the trainable length of subsequence and cluster numbers in future work. In addition, we append the specific parameters setting in line 392, lines 420 to 428, and line 501 of Section 5.
>
>
> > Please compare your work with more related studies (see Weakness 4)
>
> Thank you for the suggestion. We have updated the manuscript by appending the comparison statement with the mentioned reference in lines 85 to 96 of Section 2.
>
> Reference:
>
> [1] Azzolin, S., Longa, A., Barbiero, P., Liò, P., & Passerini, A. (2022). Global explainability of gnns via logic combination of learned concepts. arXiv preprint arXiv:2210.07147.
>
> [2] Walter, N. P., Fischer, J., & Vreeken, J. (2024, March). Finding interpretable class-specific patterns through efficient neural search. In Proceedings of the AAAI Conference on Artificial Intelligence (Vol. 38, No. 8, pp. 9062-9070).
>
> [3] Wang, B., Li, L., Nakashima, Y., & Nagahara, H. (2023). Learning bottleneck concepts in image classification. In Proceedings of the ieee/cvf conference on computer vision and pattern recognition (pp. 10962-10971).
>
> [4] Pietro Barbiero, Gabriele Ciravegna, Francesco Giannini, Pietro Lió, Marco Gori, Stefano Melacci: Entropy-Based Logic Explanations of Neural Networks. AAAI 2022: 6046-6054
>
> [5] Shan-Hwei Nienhuys-Cheng, Ronald de Wolf:
> Foundations of Inductive Logic Programming. Lecture Notes in Computer Science 1228, Springer 1997, ISBN 3-540-62927-0

---

> > ### Comment · Reviewer_iHmD · 2024-11-22
> >
> > Thank you for your responses and improvements, these are all great(except the reference to DIFFNAPS in line 92 seems to be incorrect, please correct it). I am bumping my score by 1.

---

> > > ### Author Response · Authors · 2024-11-23
> > >
> > > Thank you very much for acknowledging our responses and raising the score. We have updated the corrected reference for DIFFNAPS in line 92 in the revision. We appreciate that the questions and concerns you raised in the review are very constructive and additive to our technical contributions.

---

### Official Review · Reviewer_2bKk · 2024-11-01

**Soundness:** 2
**Presentation:** 1
**Contribution:** 2
**Rating:** 5
**Confidence:** 3

**Summary:**

The paper proposes a supervised neurosymbolic model for time series data, structured as a cascade of three main building blocks. First, an autoencoder extracts an embedding representation from raw data. Next, a differentiable clustering module maps the embedding to a symbolic representation, followed by a fuzzy neural rule learner that maps the symbolic representation to the target. The overall training objective includes three loss functions, one for each building block: a reconstruction error for the autoencoder, a k-means objective for the differentiable clustering, and a supervised cross-entropy loss for the rule learner.
Training occurs in two stages, beginning with the pre-training of the autoencoder, followed by the fine-tuning of the entire model. Additionally, a post-hoc heuristic is introduced to extract rules in propositional logic form from the weights learned by the rule learner. Experiments are conducted on a series of small datasets from UCR, comparing the model to existing neural and kernel-based classification baselines for time series data, with results demonstrating superior predictive accuracy. Lastly, a qualitative experiment on MNIST data is provided.

**Strengths:**

1. The paper is overall clear **Clarity**. However the text should be polished to rewrite/rearrange words in some sentences and correct the several typos present throughout. Please refer the MINOR paragraph below for a non-exhaustive list.
2. The considered problem is relevant and timely **Relevance**.
3. Code is available, but no further check has been performed **Code availability**.

MINOR \
L.109 -> $\neg\alpha_2$ \
L.129 -> a substitution or an interpretation \
L.151 -> is ordered real-valued observations with the dimension one \
L.159 -> discrete -> discretize \
L.161 -> closest \
L.195-197 -> could you rephrase the sentence? \
L.273 -> the its \
L.325-327 -> could you rephrase the sentence? \
L.381 -> synthetic

**Weaknesses:**

1. The paper lacks adequate context within the existing neuro-symbolic learning literature, making its contribution appear incremental in terms of novelty **Quality**. For example, neuro-symbolic (NeSy) autoencoders have been previously explored (see [1,2]), and differentiable clustering has also been investigated (refer to [3,4]). Additionally, the rule learner shows a strong resemblance to $\delta$ILP. At a minimum, a discussion of related work should be included to clarify the paper's novelty and contribution. An experimental comparison with existing methods could also help demonstrate the advantages of the proposed approach.
2. The paper lacks soundness due to several incorrect and overly ambitious claims that are not adequately supported **Soundness**. First, the paper states that this work is the first to enable joint training of the neural network and rule learner without pre-training. However, this claim is inaccurate, as the proposed training still requires a two-stage process. Additionally, contrary to what is mentioned, the work in [5] already achieves joint learning of a neural network and a rule learner. An experimental comparison on the datasets introduced in [5] would be valuable. Furthermore, the paper suggests that the proposed solution can uncover underlying rules without identifying conditions for failure. However, depending on the level of supervision and the complexity of the symbolic task, the neural learner may learn incorrect mappings from raw data to symbols. The work in [3,4] presents a basic sequential setting to analyze this issue, demonstrating that while representation learning (through differentiable clustering) is necessary to address the problem, it is not sufficient. Similar observations are subsequently reported in [6,7], where the authors label this issue as "reasoning shortcuts."
3. There is very limited discussion about the details of the experimental methodology, raising serious concerns regarding reproducibility **Reproducibility**
4. Experiments are conducted on small datasets with relatively simple symbolic tasks **Quality**.  I encourage the authors to consider more traditional and more complex experimental settings for neuro-symbolic learning, including the arithmetic tasks on handwritten digits or the datasets in [5].

**References**

[1] Misino, Marra, Sansone. VAEL: Bridging Variational Autoencoders and Probabilistic Logic Programming. In NeurIPS 2022 \
[2] Zhan, Sun, Kennedy, Yue, Chauduri. Unsupervised Learning of Neurosymbolic Encoders. In TMLR 2022 \
[3] Sansone, Manhaeve. Learning Symbolic Representations Through Joint GEnerative and DIscriminative Training. In ICLR NeSy-GeMs Workshop 2023 \
[4] Sansone, Manhaeve. Learning Symbolic Representations Through Joint GEnerative and DIscriminative Training. IJCAI KBCG Workshop 2023 \
[5] Evans et al. Making Sense of Sensory Input. In Artificial Intelligence 2021 \
[6] Marconato et al. Neuro-Symbolic Continual Learning: Knowledge, Reasoning Shortcuts and Concept Rehearsal. In ICML 2023 \
[7] Marconato, Teso, Vergari, Passerini. Not All Neuro-Symbolic Concepts Are Created Equal: Analysis and Mitigation of Reasoning Shortcuts. In NeurIPS 2023

**Questions:**

In addition to discuss the above mentioned weaknesses, please find below some additional questions:

1. The proposed solution introduces several hyperparameters, including parameters for data pre-processing (such as subsequence length and the number of temporal/spatial slots), fixed biases for the rule learner’s neurons, temperature parameters for the rule learner’s weights, and weights for the loss functions. How are these hyperparameters selected, and how sensitive is the model’s performance to their values? A sensitivity analysis could clarify this, and it should be feasible given the small size of the datasets.
2. The post-hoc strategy achieves lower predictive accuracy compared to the full neural solution (see Table 1), with drops of up to 15 percentage points in some cases. This underscores the heuristic nature of the proposed strategy. Could you provide an explanation for this phenomenon? Additionally, what can be said about the faithfulness of the extracted rules in relation to the learned model’s weights?
3. Why is neuro-symbolic learning necessary for the chosen MNIST task, and what is the rationale for this choice? The task could be solved by relying on superficial image cues (such as identifying regions that are either black or white), as illustrated by the qualitative demonstration in Figure 4, without needing to understand the underlying semantics of the digits. Therefore, a purely neural approach might suffice for this task. It would be interesting to assess the model’s capability to recognize digit classes within arithmetic tasks or, more broadly, in tasks defined by programs over integer sequences (e.g., sequences of even or odd numbers, counting).
4. Are the codes learnable in the differentiable clustering layer and how do you ensure that meaningful clusters can be learnt, without experiencing any form of collapse? Please refer for instance to the work in [1].

**References**

[1] Sansone. The Triad Of Failure Modes and a Possible Way Out. In NeurIPS SSL Workshop 2023

---

> ### Author Response · Authors · 2024-11-22
> **Responses to the Weaknesses Part 1**
>
> Thank you for your valuable review. We have included detailed responses following each of your comments.
>
> >MINOR
> >L.109 ->
> >L.129 -> a substitution or an interpretation
> >L.151 -> is ordered real-valued observations with the dimension one
> >L.159 -> discrete -> discretize
> >L.161 -> closest
> >L.195-197 -> could you rephrase the sentence?
> >L.273 -> the its
> >L.325-327 -> could you rephrase the sentence?
> >L.381 -> synthetic
>
> Thank you very much for your helpful review. We have thoroughly proofread the manuscript and corrected the typos mentioned above.
>
> >Weaknesses:
> >The paper lacks adequate context within the existing neuro-symbolic learning literature, making its contribution appear incremental in terms of novelty Quality. For example, neuro-symbolic (NeSy) autoencoders have been previously explored (see [1,2]), and differentiable clustering has also been investigated (refer to [3,4]).
>
> Thank you for the references.  We will add all these references to our related works in the revision. Indeed, we agree that our trial of adopting the autoencoder model into the neuro-symbolic research field is not the first. However, adopting autoencoder and the clustering method together on raw inputs in the rule learning research community is a novel contribution of our work. However, the research problem is different from the given reference [1,2].
>
> Misino et al. [1] used an autoencoder to logic programming field to perform label classification tasks with supporting existing domain knowledge in rules. The autoencoder and an MLP in their model calculate the probability of the given symbolic atoms as the input for ProbLog. However, the main problem in our paper is to perform inductive logic programming tasks: Learning rules as domain knowledge from raw inputs to describe the ground truth class with the features inside the raw inputs. Specifically, the autoencoder in our tasks does not learn the probabilities of a whole raw instance but learns the embeddings for the subcomponent in the raw input. A differentiable clustering method then learns discretized clusters based on these embeddings. Each cluster is regarded as a set of similar patterns and can be represented by a symbol. Then, a differentiable rule module can be used to find logic conjunctions of these clusters to describe the ground truth class. Because the whole training process is fully differentiable, the gradients computed from the rule learning module can be used to adjust clusters, which avoids cluster collapse [8].
>
> Zhan et al. [2] also used an autoencoder to learn the probability of predefined symbols speculated by a domain-specific programming language. In our setting, we wish to mine significant features as atoms in a logic program without very specific domain language. In our defined language bias, through setting the period predicate and the pattern predicate, the learned logic programs from NeurRL can describe any discriminative conjunctions of features to distinguish the ground truth class. Ultimately, our model can learn rules on multiple tasks, such as image and time series data, without modifying the language bias.
>
> For the clustering method, the main contribution of our paper is to train differentiable clustering method rules, the differentiable inductive logic programming method, and the autoencoder to learn logic programs (or logical constraints) in an end-to-end way. In this way, we can learn the logical relationships between different patterns in raw input data like time series and images. Not similar to the methods [3] and [4] proposed by Sansone and Manhaeve, the clusters are used for ProbLog for logic programming reasoning with logic program as input. Combining two lines of research, we agree that the clustering method is an excellent method to be adopted in the neuro-symbolic research community, including both deductive logic programming research and inductive logic programming research.

---

> ### Author Response · Authors · 2024-11-22
> **Responses to the Weaknesses Part 2**
>
> > Additionally, the rule learner shows a strong resemblance to $\delta$ILP. At a minimum, a discussion of related work should be included to clarify the paper's novelty and contribution. An experimental comparison with existing methods could also help demonstrate the advantages of the proposed approach.
>
> Thank you for the suggestion. We believe the $\delta$ILP in the comment is actually $\partial$ILP is a significant work in the ILP community proposed by Evans and Grefenstette [12]. We compare more differences between the rule-learning module in NeurRL with $\partial$ILP at line 75 of page 2 in the revision. Specifically, $\partial$ILP learning logic programs with all candidate rules as input, the neural networks find the weight for all these candidate rules. Then, the high probability rules are kept as the final hypothesis. Hence, the number of atoms in a rule is predefined in $\partial$ILP. Besides, because of generating candidate rules, $\partial$ILP is memory-intensive, which limits its scalability.
>
> In our proposed model, we consider the patterns of raw inputs and the regions of the patterns as atoms in the hypothesis. Motivated by the bottom clause propositionalalization method [13, 14] in the inductive logic programming theory, our model learns high probability atoms inside the hypothesis rules. Hence, our method offers better scalability and does not rely on intensive logic templates or extensive prior knowledge as input.
>
> > The paper lacks soundness due to several incorrect and overly ambitious claims that are not adequately supported Soundness. First, the paper states that this work is the first to enable joint training of the neural network and rule learner without pre-training. However, this claim is inaccurate, as the proposed training still requires a two-stage process. Additionally, contrary to what is mentioned, the work in [5] already achieves joint learning of a neural network and a rule learner. An experimental comparison on the datasets introduced in [5] would be valuable.
>
> Thank you for the thoughtful comments. We mentioned that “without pre-training” means there is no need to pre-train a large neural network in the context of label leakage [9] to transform the features of raw inputs into discretized symbols. However, the other neuro-symbolic ILP methods such as the reference [5] mentioned by the reviewer and $\alpha$ILP [10] have to use pre-trained neural networks with label information. The proposed model NeurRL supports learning symbolic rules in a fully differentiable way with the raw data as inputs and avoids label leakage. However, to improve the learning accuracy of rules, we can first train the autoencoder in an unsupervised way and obtain the clusters without leaking any labels of features or inputs, then we train the differentiable clustering method, the autoencoder, and the rule learning module to learn rules together. The gradients from the rule learning module can fine-tune the parameters in the autoencoder and the embeddings of the clusters to improve the prediction performance based on the rules. Because we only learn the embeddings for subsequences, learning clusters for these subsequences is very fast. Besides, this training strategy is a common setting that can also be used in Deep $k$-Means[11]. To avoid the misunderstanding, we change the term ‘end-to-end’ to differentiable in the revision of the manuscript.
>
> When comparing our work with the mentioned study [5], the motivations behind our model and theirs are fundamentally different. Evans et al. [5] tried to induce rules to describe the discretized series data, and each state can be interpreted as a symbolic atom. A rule learning module can only be adopted when these pre-trained neural networks are given. In our model, we target to describe continuous time series data or image data with their inside patterns without the feature engineering method. Hence, comparing two methods with different motivations needs to modify the model training strategy for the current stage. For example, the labels of two problems are different.  For describing the ground truth class, the supervised labels are the same as the classes of the instances. For performing the rule learning to describe the series transition with raw inputs, the supervised labels could be the clustering indexes of the handwritten digits. However, we regard this as future work, and we discuss it in line 538 of Section 6 in the revised manuscript.

---

> ### Author Response · Authors · 2024-11-22
> **Responses to the Weaknesses Part 3**
>
> > Furthermore, the paper suggests that the proposed solution can uncover underlying rules without identifying conditions for failure. However, depending on the level of supervision and the complexity of the symbolic task, the neural learner may learn incorrect mappings from raw data to symbols. The work in [3,4] presents a basic sequential setting to analyze this issue, demonstrating that while representation learning (through differentiable clustering) is necessary to address the problem, it is not sufficient. Similar observations are subsequently reported in [6,7], where the authors label this issue as "reasoning shortcuts."
>
> We agree that the "reasoning shortcuts" happen when the model maps the raw input into unintended semantics. In addition, this ''reasoning shortcuts'' problem usually occurs in deduction logic programming tasks with predefined logic programs by humans. In our inductive rule learning setting, we map each atom in the learned rules into a set of similar patterns in the raw inputs. We plot all patterns described by the learned rules inside the raw input in Figures 2, 3, and 4. Humans can intuitively build the intended semantics based on the patterns described by the learned rules inside the raw inputs. Besides, to prove the soundness of the learned rules and avoid learning irrelevant features to describe the ground truth class, we use precision and recall to evaluate rules based on the training raw data.
>
> > There is very limited discussion about the details of the experimental methodology, raising serious concerns regarding reproducibility Reproducibility.
> > Experiments are conducted on small datasets with relatively simple symbolic tasks Quality. I encourage the authors to consider more traditional and more complex experimental settings for neuro-symbolic learning, including the arithmetic tasks on handwritten digits or the datasets in [5].
> > References
> > [1] Misino, Marra, Sansone. VAEL: Bridging Variational Autoencoders and Probabilistic Logic Programming. In NeurIPS 2022
> > [2] Zhan, Sun, Kennedy, Yue, Chauduri. Unsupervised Learning of Neurosymbolic Encoders. In TMLR 2022
> > [3] Sansone, Manhaeve. Learning Symbolic Representations Through Joint GEnerative and DIscriminative Training. In ICLR NeSy-GeMs Workshop 2023
> > [4] Sansone, Manhaeve. Learning Symbolic Representations Through Joint GEnerative and DIscriminative Training. IJCAI KBCG Workshop 2023
> > [5] Evans et al. Making Sense of Sensory Input. In Artificial Intelligence 2021
> > [6] Marconato et al. Neuro-Symbolic Continual Learning: Knowledge, Reasoning Shortcuts and Concept Rehearsal. In ICML 2023
> > [7] Marconato, Teso, Vergari, Passerini. Not All Neuro-Symbolic Concepts Are Created Equal: Analysis and Mitigation of Reasoning Shortcuts. In NeurIPS 2023
>
> Thank you for your careful review. We consider different hyperparameter settings in Section 5.2. We appended some statements about the parameters setting description in lines 420 to 427 in Section 5.2. Please also see the specific statements about the setting of hyperparameters in the response to Question 1.
>
> Besides, the datasets we used in the paper include synthetic data, time series data, and image data. We agree that synthetic is very small because we want to prove the effectiveness of our ILP dataset on very small datasets. The statistical information of time series datasets is presented in Table 1. The maximum length of a single instance reaches 2014, and the number of instances reaches 9236. Hence, the data scale is not small in this scenario compared with the handwritten digits such as the MNIST data with a length of 784 (28 $\times$ 28). From the architecture of our rule-learning neural network, the number of training parameters is only related to the hyperparameters of the number of periods and clusters. Because these hyperparameters are small, our model is scalable compared with $\partial$ILP [12].
>
> We also appreciate the suggestion above for the experiment setting based on the handwritten digits defined in the task proposed by Evans et al. [5]. However, we highlight that because the problem motivation is different between our model and the work [5]. We try to use learned patterns in raw input automatically to describe the ground truth class of the raw inputs. The datasets in [5] are very discretized point series, and each data point can be mapped into a symbol atom. The model in [5] learns rules to explain the symbol transitions. We use handwritten digits such as MNIST as a benchmark because we can use pattern information and region information of subareas inside images to describe the ground truth class from the viewpoint of the global explanation. The experimental results can be understood intuitively by humans to distinguish ground truth class and negative class inputs.

---

> ### Author Response · Authors · 2024-11-22
> **Responses to the Questions Part 1**
>
> >Questions:
> >In addition to discuss the above mentioned weaknesses, please find below some additional questions:
> >The proposed solution introduces several hyperparameters, including parameters for data pre-processing (such as subsequence length and the number of temporal/spatial slots), fixed biases for the rule learner’s neurons, temperature parameters for the rule learner’s weights, and weights for the loss functions. How are these hyperparameters selected, and how sensitive is the model’s performance to their values? A sensitivity analysis could clarify this, and it should be feasible given the small size of the datasets.
>
> Thank you for the comments. We elaborate on the strategy for choosing the hyperparameter in line 392, lines 420 to 428, and line 501 in Section 5. Specifically, three essential hyperparameters are considered: the number of clusters, the length of the subsequence, and the length of the period. The hyperparameter of fixed bias is set to 0.5 in the rule learning module among all tasks. The weights $\lambda_1$ and $\lambda_2$ in the loss function are set to 1, which indicates that we assigned equal weights for finding representations, identifying clusters, and discovering rules. We add notes to explain the above parameters setting on pages 6 and 7.
>
>
> The length of the period depends on the number of periods given the fixed length of the series data. The number of clusters determines the number of atoms in the hypothesis. The length of the subsequence is related to the potential pattern’s length in the raw inputs. The number of periods and clusters affect the rule-learning module size, which can be reviewed from lines 298 to 300.
>
> To accommodate different tasks, we adjust the hyperparameters for UCR tasks and MNIST data accordingly. For the number of clusters, we set it to three for synthetic data, five for the UCR archive, and five for image datasets. For the length of the subsequence, we set it to five for synthetic data. In UCR data, the length of the subsequence ranges from two to five, depending on the task. For MNIST data, the subsequence length is set to three. Regarding the length of the period, we set it to five for the synthetic dataset. For UCR datasets, given the varying time series lengths across tasks, we generally constrain the number of periods to 10. In MNIST datasets, the length of a single period is set to three. In conclusion, the choice of hyperparameters depends on the desired explainability granularity and the varying lengths of sequence data.
>
> >The post-hoc strategy achieves lower predictive accuracy compared to the full neural solution (see Table 1), with drops of up to 15 percentage points in some cases. This underscores the heuristic nature of the proposed strategy. Could you provide an explanation for this phenomenon? Additionally, what can be said about the faithfulness of the extracted rules in relation to the learned model’s weights?
>
> For the faithfulness between rules and neural networks, the extracted rules correspond to the binarized softmax-activated weights of the rule learning module under all thresholds from 0 to 1. The post-hoc strategy uses rules to describe the ground truth class by verifying whether all patterns specified in the rule occur in a given raw input. Since rules use discrete features to predict the labels of continuous raw inputs, this may result in the prediction performance of the post-hoc strategy being lower than that of neural networks. Specifically, the raw inputs are noisy, but the patterns stipulated by rules are discrete. Some outliers in the raw inputs may cause the pattern to be different from the patterns stipulated in a learned rule. In addition, the current results accurately represent the ground truth classes in certain tasks, as shown in Table 1. This indicates that the rules capture precise and abundant patterns to describe the ground truth classes effectively. On the other hand, neural networks may learn an excessive number of patterns, potentially leading to false predictions. As a result, their performance may be lower than that achieved by rule-based predictions.

---

> ### Author Response · Authors · 2024-11-22
> **Responses to the Questions Part 2 and Reference**
>
> >Why is neuro-symbolic learning necessary for the chosen MNIST task, and what is the rationale for this choice? The task could be solved by relying on superficial image cues (such as identifying regions that are either black or white), as illustrated by the qualitative demonstration in Figure 4, without needing to understand the underlying semantics of the digits. Therefore, a purely neural approach might suffice for this task. It would be interesting to assess the model’s capability to recognize digit classes within arithmetic tasks or, more broadly, in tasks defined by programs over integer sequences (e.g., sequences of even or odd numbers, counting).
>
> We agree that a purely neural model can predict the positive class given a negative class.
> However, the main topic and contribution of our paper is to learn rules to describe the positive class given the negative class with features inside all inputs. Compared to purely neural models, they cannot provide a clear explanation using feature conjunctions to justify how a positive prediction is made.
>
> We choose MNIST images as benchmarks because we can use pattern information (black pixel or white pixel) and clear region information to describe the ground truth class.
> These explainable results in Figure 4 are very intuitive for distinguishing the positive class from the negative class.
>
> In our paper, we're learning rules with patterns inside an input as an atom. Learning rules over integer sequences or understanding the semantics of digits needs the whole input as an atom. Hence, our task is very different from the problem statement defined by [5]. Our problem, which is learning rules with patterns inside an input, has multiple practical application domains such as medical image analysis, industrial time series data analysis, data explainability, etc.
>
>
> >Are the codes learnable in the differentiable clustering layer and how do you ensure that meaningful clusters can be learnt, without experiencing any form of collapse? Please refer for instance to the work in [1].
> >References
> >[1] Emanuele Sansone. The Triad Of Failure Modes and a Possible Way Out. In NeurIPS SSL Workshop 2023
>
> We understand that collapse includes two cases in [8]. The first case is that the encoder maps every input to the same embedding. The second case is that the predictive model assigns all samples to the same cluster with high confidence. To address the issue of all subsequences being encoded into the same embedding or different embeddings being assigned to the same cluster, the rule-learning module is added after the autoencoder and differentiable clustering method. The gradients generated from the rule learning module can further adjust the embeddings generated from the autoencoder and the assignments of clusters for the embeddings. Hence, the fully differentiable loss function defined in Eq. (10) can avoid the two forms of collapses.
>
>
>
> Reference:
>
> [1] Misino, Marra, Sansone. VAEL: Bridging Variational Autoencoders and Probabilistic Logic Programming. In NeurIPS 2022
>
> [2] Eric Zhan, Jennifer J. Sun, Ann Kennedy, Yisong Yue, Swarat Chaudhuri: Unsupervised Learning of Neurosymbolic Encoders. Trans. Mach. Learn. Res. 2022 (2022)
>
> [3] Sansone, Manhaeve. Learning Symbolic Representations Through Joint GEnerative and DIscriminative Training. In ICLR NeSy-GeMs Workshop 2023
>
> [4] Sansone, Manhaeve. Learning Symbolic Representations Through Joint GEnerative and DIscriminative Training. IJCAI KBCG Workshop 2023
>
> [5] Evans et al. Making Sense of Sensory Input. In Artificial Intelligence 2021
>
> [6] Marconato et al. Neuro-Symbolic Continual Learning: Knowledge, Reasoning Shortcuts and Concept Rehearsal. In ICML 2023
>
> [7] Marconato, Teso, Vergari, Passerini. Not All Neuro-Symbolic Concepts Are Created Equal: Analysis and Mitigation of Reasoning Shortcuts. In NeurIPS 2023
>
>
> [8] Emanuele Sansone. The Triad Of Failure Modes and a Possible Way Out. In NeurIPS SSL Workshop 2023
>
> [9] Sever Topan, David Rolnick, Xujie Si: Techniques for Symbol Grounding with SATNet. NeurIPS 2021: 20733-20744.
>
> [10] Hikaru Shindo, Viktor Pfanschilling, Devendra Singh Dhami, Kristian Kersting: αILP: thinking visual scenes as differentiable logic programs. Mach. Learn. 112(5): 1465-1497 (2023).
>
> [11] Maziar Moradi Fard, Thibaut Thonet, Éric Gaussier: Deep k-Means: Jointly clustering with k-Means and learning representations. Pattern Recognit. Lett. 138: 185-192 (2020).
>
> [12] Richard Evans, Edward Grefenstette: Learning Explanatory Rules from Noisy Data. J. Artif. Intell. Res. 61: 1-64 (2018).
>
> [13] Manoel V. M. França, Gerson Zaverucha, Artur S. d'Avila Garcez: Fast relational learning using bottom clause propositionalization with artificial neural networks. Mach. Learn. 94(1): 81-104 (2014).
>
> [14] Kun Gao, Katsumi Inoue, Yongzhi Cao, Hanpin Wang: A differentiable first-order rule learner for inductive logic programming. Artif. Intell. 331: 104108 (2024).

---

> ### Author Response · Authors · 2024-11-25
> **Additional Responses and Experimental Results on the Sensitivity and Faithfulness of NeurRL as Discussed in Question 1**
>
> We hope this response addresses your constructive suggestions.
>
> In the latest revision, we have added the selection of the hyperparameter $\alpha = 1000$ for the clustering method in our experiments on page 5. Additionally, we have included more experiments to assess the sensitivity of NeurRL and added a new Section 5.4 to discuss the model’s sensitivity.
>
> Based on the results, we observe from Figure 5 (line 500) that NeurRL’s sensitivity varies across tasks. Moreover, the subsequence length is a sensitive hyperparameter when the sequence length is small, as seen in the ItalyPow.Dem. dataset.
>
> Furthermore, properly chosen hyperparameters can achieve high consistency in accuracy between rules and neural networks. Notably, increasing the number of clusters and the length of regions does not reduce accuracy linearly, suggesting that the rule-learning module effectively adjusts clusters within the model.
>
> We welcome further discussion if there are additional comments. Thank you!

---

> > ### Comment · Reviewer_2bKk · 2024-11-25
> > **Answer to Rebuttal**
> >
> > Dear authors,
> >
> > Thank you for the thorough answers and the effort made to modify the article. I appreciate the discussion about how your solution contextualises with respect to existing work (though, I’m not able to refer to some changes mentioned at specific lines in your answer, like L392, L398 to 400 L420 to 427), the clarification of using an unsupervised pre-training instead of a supervised one to avoid the symbol leakage problem (i.e. avoiding the use of explicit label supervision), the modification of the title to reflect this and the additional experiments about the sensitivity analysis of the hyperparameters.
> >
> > However, there are still a number of issues that I’m concerned with (all points are related with the Quality/Soundness of the paper):
> > 1. Regarding the claim that the proposed method can avoid collapses. I’m not sure about this statement. This seems mitigated partly by the assumption of having an unsupervised pre-training stage. What does it happen if you remove this pre-training stage? And how can you guarantee that you identify the correct patterns out of data, even when having pre-training? Also, it should be made more explicit in the abstract and introduction that you still require a pre-training stage.
> > 2. Another way to look at the above point is to provide a comparison against existing neuro-symbolic approaches where the rule are given and see whether both symbols and rules are recovered correctly. However, such analysis is currently missing in the paper.
> > 3. Regarding the procedure to extract rules from the fuzzy rule learning system. I’m not convinced about the answer for the following reasons. There is still a large performance gap between the solution and the “discretized version”. Moreover, I don’t clearly see what you mean by **neural networks may learn an excessive number of patterns, potentially leading to false predictions**. The deep rule learner (neural network solution) achieves the best performance in Table 1 while the discretised version falls short, suggesting that the rule extraction strategy is not able to faithfully capture the underlying rules. This is clearly due to the “fuzziness” of the proposed solution. Another question taking another different angle at this is: can the proposed neural strategy be really considered a differentiable rule learner?
> > 4. It is mentioned several times that the proposed strategy is scalable, but all experiments are small scale.
> >
> > Overall, I think that there has been an improvement over the previous version of the paper and I can therefore increase my score. However, I will increase it of 1 point as the above-mentioned concerns are still unaddressed.

---

> > > ### Author Response · Authors · 2024-11-27
> > > **Response to the Second-Round Comments – Part 1**
> > >
> > > We thank you for your additional suggestions and comments. We also appreciate the adjustment of the score based on the major revisions. Please find our responses to the new comments below. We are happy to discuss further if the response does not fully address your question or if you have additional comments.
> > >
> > > > Dear authors,
> > > > Thank you for the thorough answers and the effort made to modify the article. I appreciate the discussion about how your solution contextualises with respect to existing work (though, I’m not able to refer to some changes mentioned at specific lines in your answer, like L392, L398 to 400 L420 to 427), the clarification of using an unsupervised pre-training instead of a supervised one to avoid the symbol leakage problem (i.e. avoiding the use of explicit label supervision), the modification of the title to reflect this and the additional experiments about the sensitivity analysis of the hyperparameters.
> > >
> > > We apologize for not updating the line numbers in our previous response, as the manuscript was revised following the addition of the new Section 5.4. In the latest revision, we have added statements regarding the hyperparameter settings for synthetic data, UCR time series data, and MNIST data, which can now be found on line 392, lines 409–412, and line 473, respectively.
> > >
> > > > However, there are still a number of issues that I’m concerned with (all points are related with the Quality/Soundness of the paper):
> > > > Regarding the claim that the proposed method can avoid collapses. I’m not sure about this statement. This seems mitigated partly by the assumption of having an unsupervised pre-training stage. What does it happen if you remove this pre-training stage?
> > >
> > > Per-training autoencoder helps the ongoing training process to quickly find the conjunction of clustering indexes that have a logical correlation between the label of instance. In other words, the rule-learning module just fine-tuning the cluster's representations and embeddings at the same time.
> > >
> > > We present experimental comparisons with and without pre-training in Appendix A of the supplement. The table is presented as follows:
> > >
> > > | Dataset       | With pre-training |                  | Without pre-training |                  |
> > > | ------------- | :---------------: | ---------------: | :------------------: | ---------------: |
> > > |               |     Accuracy      | Running time (s) |       Accuracy       | Running time (s) |
> > > | Coffee        |    1.00,  0.96    |               42 |     0.83,  0.81      |               30 |
> > > | ECG           |    0.88,  0.82    |               65 |     0.87,  0.64      |               53 |
> > > | ItalyPow.Dem. |    0.92,  0.93    |               63 |     0.75,  0.80      |               61 |
> > > | Gun Point     |    0.87,  0.76    |               35 |     0.86,  0.43      |               31 |
> > > | Lighting2     |    0.75,  0.69    |              120 |      0.64, 0.60      |               63 |
> > >
> > > The results show that the running time with and without pre-training the autoencoder does not exhibit a large gap, as pre-training only applies to the autoencoder with MLP as the encoder and decoder. However, without the pre-training stage, both the neural network and the rule performance suffer due to the collapse situation.
> > >
> > > > And how can you guarantee that you identify the correct patterns out of data, even when having pre-training?
> > >
> > > The accuracy of neural networks and rules can reflect whether the collapses happen. If all patterns are assigned to one cluster, whatever the failures happen when learning embeddings of patterns or learning clusters, then the rule-learning module can return the rule with that one atom as the body. Then, the accuracy of the rule and neural networks will always be the ratio of the number of positive examples to all examples. Hence, good accuracy can prove that the rule-learning module adjusts the clusters and embeddings. Besides, we can even plot the patterns in rules with the raw inputs, and we can see these highlighted patterns are obvious to distinguish positive examples from negative examples. Furthermore, we calculate the precision and recall based on the clusters of subsequences of rules. Hence, the plotted results, precision, and recall also can prove the system can learn correct knowledge from data.
> > >
> > > > Also, it should be made more explicit in the abstract and introduction that you still require a pre-training stage.
> > >
> > > Requiring a pre-training stage does not alter our main contribution, which is the ability to train the model without using explicit input feature labels. Therefore, in the abstract, we maintain our emphasis on not needing explicit supervision labels for input features. However, we provide further clarification on how we avoid explicit labels by including a statement about the unsupervised pre-training stage in line 53 of the revised introduction.

---

> > > ### Author Response · Authors · 2024-11-27
> > > **Response to the Second-Round Comments – Part 2**
> > >
> > > > Another way to look at the above point is to provide a comparison against existing neuro-symbolic approaches where the rule are given and see whether both symbols and rules are recovered correctly. However, such analysis is currently missing in the paper.
> > >
> > > We understand the question and agree that pre-setting rules and checking whether the model can retrieve the correct rules is an effective way to assess performance. However, we believe that Section 5.1 can already prove the model can retrieve the pre-defined rules from data.
> > >
> > > In this section, we manually define positive and negative examples with specific patterns based on pre-defined rules (or constraints) to test the model’s ability to retrieve these rules. The constraint is as follows: there are three distinct regions in both positive and negative examples, with different patterns in each region for positive and negative examples. For instance, in the second region of Figure 2 (a), the pattern is an increment for negative examples and a decrement for positive examples.
> > >
> > > The model is expected to make the correct classification prediction by learning a rule with any one of the regions and the corresponding pattern as specified by our constraints. The results in Figure 2 convincingly demonstrate that NeurRL can accurately output the predefined rules from the sequence data.
> > >
> > > > Regarding the procedure to extract rules from the fuzzy rule learning system. I’m not convinced about the answer for the following reasons. There is still a large performance gap between the solution and the “discretized version”.
> > >
> > > The performace gap between the neural networks and logic rules is different on different tasks in the UCR archive in Table 1.  In some situations, the gap is less than 0.005. The difference between rules and neural networks is less than 0.05 in 9 tasks of all 13 tasks.
> > >
> > > The lower prediction accuracy of rules compared with the neural network module happens because of the discretization. The prediction accuracy by rules is made after the discretization of the clustering results and the weights of the rule-learning module. Hence, the continuous output of the clustering method and the continuous rule embeddings may have better prediction accuracy in general.
> > >
> > > However, the discretized rules extracted from neural networks and the clustering index of patterns build the interpretability of the NeurRL. These highlighted patterns corresponding to the rules in the raw data help people to know the main knowledge hidden in the data. Besides, the classification performance of rules extracted from neural networks is still comparable with other SOTA models.
> > >
> > > > Moreover, I don’t clearly see what you mean by neural networks may learn an excessive number of patterns, potentially leading to false predictions.
> > >
> > > Sorry for the confusion. In ItalyPow.Dem. dataset, the prediction accuracy of rules is higher than the neural networks under some hyperparameter settings. This situation can happen in the situation when the extracted rules precisely reflect the patterns and the patterns are not ambiguous to mislead the rules. However, the weights in the neural networks continue and include other learned patterns, and the cluster indexes are also not binaries. Hence, the prediction of the neural network may be infected. This situation also happens in other differentiable rule learning methods such as [7].
> > >
> > > >The deep rule learner (neural network solution) achieves the best performance in Table 1 while the discretized version falls short, suggesting that the rule extraction strategy is not able to faithfully capture the underlying rules. This is clearly due to the “fuzziness” of the proposed solution.
> > >
> > > In NeurRL, all the possible ground atoms would be the body of a rule before training the neural networks. Based on the logic programming process in algebra space, we design the neural network forwarding computation process. Through the supervised training on the rule-learning module, the well-trained weights correspond to the logic program matrix $M_P$ defined in Eq. (1), and the atoms corresponding to the high possibility weights are regarded as the body of rules.
> > >
> > > Based on the fact that the atoms correspond to the weights of the high values as the learned rule body, we design the rule extraction by setting a series of thresholds. The atoms whose weights are larger than the threshold would be the body atoms of rules. In this way, **all possible** learned rules will be extracted in linear time complexity. Then,  we check the precision and recall of potential rules, and these rules with high precision and recall values would be the final output. The rule extraction method based on the design motivations of neural networks is common in neuro-symbolic inductive logic programming research [1,2,5].

---

> > > ### Author Response · Authors · 2024-11-27
> > > **Response to the Second-Round Comments – Part 3 and Reference**
> > >
> > > > Another question taking another different angle at this is: can the proposed neural strategy be really considered a differentiable rule learner?
> > >
> > > Our proposed method, NeurRL, is a differentiable rule learner. Similar to NeurRL, other differentiable rule learners, such as those by França et al. [1], Yang et al. [2], and Sadeghian et al. [5], design white-box neural networks to perform classification tasks and then extract rules from the well-trained networks. It is important to note that the method of rule extraction is closely tied to the architecture of the neural networks. Therefore, these differentiable rule learners do not learn the weights [3] for filling the gaps in logic templates [4], but rather focus on learning the structure of logical rules.
> > >
> > >
> > > Beyond the domain of inductive logic programming, the KAN proposed by Liu et al. [6] also extracts mathematical equations from well-trained neural networks, which can be considered a differentiable model for discovering mathematical laws.
> > >
> > >
> > >
> > > > It is mentioned several times that the proposed strategy is scalable, but all experiments are small-scale.
> > >
> > > From the perspective of the scalability of the rule-learning module, we extend the model based on [1]. In this paper, scalability is demonstrated using knowledge graph data. Theoretically, scalability depends on the structure of the neural networks and how the neural network builds the rules. NeurRL builds the logical structure of rules and does not rely on the time-consuming rule clause-generating algorithm. Besides, the input node of the neural network is related to the number of clusters and the number of regions. Hence, the neural network can be trained easily.
> > >
> > > From a practical standpoint, the benchmarks include both UCR time series data, which covers a wide range of real-world scenarios such as medical (ECG) and astronomy (StarLightCurves), as well as small synthetic datasets. Thus, our model can be used in these realistic cases. Based on these statements, we assert that our model is scalable.
> > >
> > >
> > >
> > > Reference:
> > >
> > > [1] Manoel Vitor Macedo França, Artur S. d'Avila Garcez, Gerson Zaverucha: Relational Knowledge Extraction from Neural Networks. CoCo@NIPS 2015.
> > >
> > > [2] Fan Yang, Zhilin Yang, William W. Cohen: Differentiable Learning of Logical Rules for Knowledge Base Reasoning. NIPS 2017: 2319-2328.
> > >
> > > [3] Richard Evans, Edward Grefenstette: Learning Explanatory Rules from Noisy Data. J. Artif. Intell. Res. 61: 1-64 (2018).
> > >
> > > [4] Robin Manhaeve, Sebastijan Dumancic, Angelika Kimmig, Thomas Demeester, Luc De Raedt: Neural probabilistic logic programming in DeepProbLog. Artif. Intell. 298: 103504 (2021).
> > >
> > > [5] Ali Sadeghian, Mohammadreza Armandpour, Patrick Ding, Daisy Zhe Wang: DRUM: End-To-End Differentiable Rule Mining On Knowledge Graphs. NeurIPS 2019: 15321-15331.
> > >
> > > [6] Ziming Liu, Yixuan Wang, Sachin Vaidya, Fabian Ruehle, James Halverson, Marin Soljacic, Thomas Y. Hou, Max Tegmark: KAN: Kolmogorov-Arnold Networks. CoRR abs/2404.19756 (2024).
> > >
> > > [7] Artur S. d'Avila Garcez, Krysia Broda, Dov M. Gabbay: Symbolic knowledge extraction from trained neural networks: A sound approach. Artif. Intell. 125(1-2): 155-207 (2001).

---

> > > > ### Comment · Reviewer_2bKk · 2024-12-01
> > > > **Thank You**
> > > >
> > > > Thank you for the additional clarifications (for adjusting the numbering and providing experiments without pre-training).
> > > > I still think that the concerns highlighted in the last answer are not addressed, especially about collapses, comparison with supervised neuro-symbolic baselines, faithfulness of the rules and scalability of the experiments. Based on this I keep my score.

---

### Official Review · Reviewer_K7Gv · 2024-11-02

**Soundness:** 3
**Presentation:** 3
**Contribution:** 2
**Rating:** 6
**Confidence:** 4

**Summary:**

The paper proposes a new differentiable Inductive Logic Programming framework where symbolic rules are learned via gradients from sequential data. To overcome the bottleneck of previous models that require pre-trained neural networks to perceive raw input, the authors propose Neural Rule Learner (NeurRL), which combines VAE and differentiable k-means to the differentiable rule learning approach. The resulting framework can learn classification rules that specify regions and corresponding patterns to be classified as positive. It is demonstrated in the experiments using simple datasets, including synthetic, UCR, and MNIST datasets.

**Strengths:**

This paper tackles an important topic in the field of differentiable ILP, specifically the need for pre-trained models to ground raw input into symbols for effective rule learning. The approach of combining differentiable k-means and VAE with gradient-based rule learners is noteworthy, even though the concept is relatively straightforward.

The manuscript is well-written, with a high-quality presentation overall. It clearly outlines the research question and effectively conveys its core idea.

This work is a valuable contribution to the neuro-symbolic research community, paving the way for developing more robust systems capable of learning with fewer priors on raw input.

**Weaknesses:**

My primary concern is that it remains unclear whether the main claim is adequately supported by the methods and experiments sections. The paper asserts:

> (Line 42-45) Learning logic programs from raw data is hindered by the label leakage problem common in neuro-symbolic research (Topan et al., 2021): This leakage occurs when labels of ground objects are introduced for inducing rules (Evans & Grefenstette, 2018; Shindo et al., 2023).

and

> (Line 82-85) In our work, we do not require a pre-trained large-scale neural network to map raw input data to symbolic labels or atoms. Instead, we design an end-to-end learning framework to derive rules directly from raw numeric data, such as sequence and image data.

If this claim holds, the proposed system should be trainable end-to-end, grounding symbols in meaningful ways such that rule learners can generate rules based on them. However, the paper does not make it clear how this grounding is accomplished.

Does NeurRL perform grounding on the obtained clusters? Can the rule learners manage rules involving multiple clusters by understanding the meaning of each cluster? If not, I am skeptical that the proposed framework fully addresses the bottleneck identified in the introduction, namely the reliance on perception models in prior rule learning frameworks like $\partial$ ILP and $\alpha$ ILP.

Without grounding, the method would be confined to the rule format (Eq. 5), as the system would lack the means to learn from non-grounded symbols, limiting its applicability to other domains.

While I appreciate the method's ability to learn from small datasets, it is unclear how it would scale to larger, more complex datasets. Given that neural models typically require large datasets, future development would benefit from the capacity to handle such data at scale to better integrate neural networks. It would be beneficial to provide runtime comparisons on different dataset sizes, or to discuss potential challenges and solutions for scaling the approach to larger datasets. How does the model's performance change with increasing dataset complexity and size? If scalability to large-scale data is an issue, this limitation should be discussed somewhere in the paper.

**Questions:**

Differentiable ILP frameworks ($\partial$ ILP and $\alpha$ ILP) typically suffer from their intense memory consumption due to the tensor encoding. Does NeurRL share the same problem?

For other questions, please refer to the weakness section.

---

> ### Author Response · Authors · 2024-11-22
> **Responses to the Weaknesses Part 1**
>
> Thank you for the thoughtful review. Please see our response in the following.
>
> >Weaknesses:
> >My primary concern is that it remains unclear whether the main claim is adequately supported by the methods and experiments sections. The paper asserts:
> >(Line 42-45) Learning logic programs from raw data is hindered by the label leakage problem common in neuro-symbolic research (Topan et al., 2021): This leakage occurs when labels of ground objects are introduced for inducing rules (Evans & Grefenstette, 2018; Shindo et al., 2023).
> >and
> >(Line 82-85) In our work, we do not require a pre-trained large-scale neural network to map raw input data to symbolic labels or atoms. Instead, we design an end-to-end learning framework to derive rules directly from raw numeric data, such as sequence and image data.
> >If this claim holds, the proposed system should be trainable end-to-end, grounding symbols in meaningful ways such that rule learners can generate rules based on them. However, the paper does not make it clear how this grounding is accomplished.
> >Does NeurRL perform grounding on the obtained clusters? Can the rule learners manage rules involving multiple clusters by understanding the meaning of each cluster? If not, I am skeptical that the proposed framework fully addresses the bottleneck identified in the introduction, namely the reliance on perception models in prior rule learning frameworks like $\partial$ILP and $\alpha$ILP.
> >Without grounding, the method would be confined to the rule format (Eq. 5), as the system would lack the means to learn from non-grounded symbols, limiting its applicability to other domains.
>
> In our paper, the training process is a fully differentiable process from the raw data to symbolic rules without leaking labels for patterns in raw inputs. Yes, the clustering method can assign a symbol atom to one cluster because the subsequences in one cluster are very similar. However, this symbol atom is abstract and does not describe the semantics of similar patterns in one cluster. Then, the rule learning module can find the highly related conjunctions of these clusters with the ground truth class. Consequently, we present the rules and its original raw input data to illustrate the learned knowledge. Hence, in this process, training the whole model only needs the label of the raw inputs without a pre-trained perception to recognize all objects in the input and return their label.
>
> The symbol grounding problem [1] is defined as the inability to map raw inputs to symbolic variables without explicit supervision ('label leakage'). Both $\partial$ILP and $\alpha$ILP need a well-trained perception to transfer raw inputs to correct semantics symbolic labels first and then use ILP methods to induce rules or describe the image with rules with symbols in pre-defined labels. Hence, compared with these state-of-the-art ILP models, we avoid the sense of label leakage to induce rules to explain the ground truth class.
>
> Even though the learned rules can be described by the rule in Eq. (5), the rule in Eq. (5) includes an arbitrary feature represented by the predicate $pattern_i$. We can present the learned features by plotting the feature in the original raw inputs to achieve explainability.
>
> Similarly, the learning process of NeurRL is closed with the motivated paper [1]. In the clustering method, even though we can aggregate similar subsequences into one cluster, we still have no explicit labels or information to describe each cluster in the context of avoiding label leakage. The viable approach to establishing a logical relationship between these clusters and the ground-truth class is through the differentiable rule learner. This process identifies the correct rules using these clusters as body atoms and adjusts the clustering process and encoding process accordingly. Topan et al. [1] also mentioned that they cannot explicitly return the unsupported labels for all vision input cells with MNIST images but learned a permutation matrix between real labels and cluster indexes of the MNIST handwritten digits.
>
>
> Reference:
>
> [1] Sever Topan, David Rolnick, Xujie Si: Techniques for Symbol Grounding with SATNet. NeurIPS 2021: 20733-20744

---

> ### Author Response · Authors · 2024-11-22
> **Responses to the Weaknesses Part 2 and Questions**
>
> >While I appreciate the method's ability to learn from small datasets, it is unclear how it would scale to larger, more complex datasets. Given that neural models typically require large datasets, future development would benefit from the capacity to handle such data at scale to better integrate neural networks. It would be beneficial to provide runtime comparisons on different dataset sizes, or to discuss potential challenges and solutions for scaling the approach to larger datasets. How does the model's performance change with increasing dataset complexity and size? If scalability to large-scale data is an issue, this limitation should be discussed somewhere in the paper.
>
> In our experiment, we run the model on synthetic datasets, which are very small, and the positive and negative classes both include two instances. Hence, NeurRL can solve smaller datasets. For the larger datasets, we listed the data statistical information in Table 1. The largest used data in the UCR archive is StarLightCureves, which includes 9236 instances, and the length of each instance is 2014. We also presented some running time results in Table 2. Currently, we can obtain results on all UCR tasks within 10 minutes of training time on NVIDIA GeForce RTX 3090. The running time is fast because we run the differentiable clustering method on the subsequence of the time series. In addition, the neural network size of the rule-learning module is related to the number of periods and number of clusters. Assume the number of periods and the number of clusters to $P$ and $K$, respectively. Then, the input node of the rule-learning module would be $P\times K$. For most tasks, we set the number of periods to 10 and set the number of clusters to 5. Hence, the number of parameters of the rule-learning neural network is quite small. Hence, the scalability of the model is an advantage of the model.
>
> >Questions:
> >Differentiable ILP frameworks ( $\partial$ILP and $\alpha$ILP) typically suffer from their intense memory consumption due to the tensor encoding. Does NeurRL share the same problem?
> >For other questions, please refer to the weakness section.
>
> The $\partial$ILP and $\alpha$ILP suffered intense memory because the logic clauses in these ILP systems are pre-generated. Specifically, logic templates are used in $\partial$ILP, and top-$k$ beam search algorithm is used in $\alpha$ILP. The neural network is used only to train the weights of these pre-generated candidate clauses in $\partial$ILP and $\alpha$ILP. Hence, it's hard to fully utilize GPU computation resources to obtain rules for $\partial$ILP and $\alpha$ILP.
>
> In our model NeurRL, we design a fully differentiable neural network structure for learning rules. The main motivation of our rule-learning module is to find the optimal atoms to form a rule but not find the optimal weights based on the well-generated rule candidates. Besides, the size of the neural network is very small, because the input node is $P\times K$, the number of nodes in our two hidden layers and output layer are 2, 2, and 1, respectively. Hence, adopting a neural network solely to find rules makes our work more scalable and avoids running out of memory errors.

---

> ### Comment · Reviewer_K7Gv · 2024-11-25
> **Response to Authors**
>
> I thank the authors for their clarifications. My concerns have been partially addressed.
>
> >  training the whole model only needs the label of the raw inputs without a pre-trained perception to recognize all objects in the input and return their label.
>
> This sounds great, but is it guaranteed? Would it produce a reasonable solution if the training data is noisy and not well-separated to form clear clusters? The conventional rule learners ($\partial$ILP and $\alpha$ILP) indeed rely on pre-trained perception models. This implies that they can take any perception model depending on the dataset in interest (this would not be critical as NeurRL could also incorporate any pre-trained models).
> My point is more about the capability and limitations of the proposed approach. As the perception models have been widely studied, it is somewhat easy to expect their applicability.
>
>
> This is related to the answer:
> > We can present the learned features by plotting the feature in the original raw inputs to achieve explainability.
>
> I think this statement holds only when neural networks (or data features) can produce good explanations. Any discussion on assumptions and limitations (or even showcasing failure cases) regarding the proposed method would help future readers identify the scope of the paper and the applicability of the proposed method.
>
> Overall, I will maintain my rating given the concerns above. I appreciate the contribution to reducing the amount of supervision for differentiable rule learners.
>
>
> Minor:
> Table 2 does not articulate the unit (second?), and I suggest clarifying it in the table.

---

> > ### Author Response · Authors · 2024-11-25
> >
> > We thank you for your constructive feedback and suggestions regarding the unit in Table 2. We have modified Table 2 in the revised version.
> >
> > When the input raw data contains many missing values, the clustering method may assign subsequences based only on the mean value, as per the current method. As a result, the pattern information may not be sufficient to capture knowledge from such data with missing values. Therefore, learning knowledge to describe sequence data with a large number of missing values is a direction for future work. We have added this statement to Conclusion in the revision.

---

### Official Review · Reviewer_6MqZ · 2024-11-06

**Soundness:** 3
**Presentation:** 4
**Contribution:** 3
**Rating:** 8
**Confidence:** 3

**Summary:**

This work introduces Neural Rule Learner (NeurRL), a system designed to learn interpretable rules from sequential data, such as from time-series sensor data (like spectrograms) and serialized higher-dimensional data (such as images).

The system uses symbolization functions to map raw data into a predetermined framework of pattern and region predicates, allowing discrete raw data sequences to be represented as logical atoms that can be used in rule learning. In the supplied reference implementation, patterns are learned automatically through a differentiable k-means clustering process, while regions are predefined subdivisions of the sequence. The combination of learned patterns and predefined regions allows the system to express discovered features as logical rules using this symbolic vocabulary.

The framework uses two types of symbolization functions: body symbolization functions that map discrete sequences to symbolic concepts, and head symbolization functions that map inputs to a set of atoms. The system avoids the "label leakage" problem by not requiring pre-trained neural networks to map raw inputs to symbolic labels. The entire pipeline is differentiable and can be trained end-to-end, learning directly from raw data without needing intermediate symbolic representations.

The implementation uses a deep rule-learning module with multiple dense layers, and rules are evaluated using precision and recall metrics. The reference implementation includes an autoencoder component for learning concentrated representations of subsequences.
Experimental validation was conducted on both time series data (UCR datasets) and image data (MNIST).  Learning on UCR data-subsets  highlights low-data or data-efficient learning methods. The reference implementation achieves comparable or better classification accuracy compared to baseline methods, and provides interpretable rules. The generated rules are human-readable and capture patterns in the data.

**Strengths:**

Originality: This provides a new and instructive implementation to address the raw-to-symbolic data problem, while avoiding the data leakage issue.

Quality and Clarity: The paper explains and illustrates the method in a clear and compelling way.

Significance:  This appears to be an important contribution to the problem of addressing the problem of learning symbolic rules about raw sequential data.

**Weaknesses:**

The approach may require significant data invariance (orientation of images, sampling regularity).  Many important datasets have these properties, although image data often does not.  See questions for further clarification.

As a demonstration of the advantage of this approach as an interpretable and explainable method, it would be helpful to have the interpretation of rules on each dataset from UCR discussed in more detail.  How interpretable are the rules in terms of the data patterns?  Are there particular datasets or data types for which rules emerge that provide insight or match intuition?

**Questions:**

Question:  Please clarify the generalizability of your approach.  Do you expect the rule learning method to be sensitive to data errors that arise naturally in real-world collection, such as uncalibrated sensors or aspect/focus variation in images, or frame rate in video?   Consider the following comment, if that helps clarify the objective of the request.

I am not familiar with the UCR datasets, and I don't think you provided a link, nor did the paper you cite have a currently valid link; I found data and descriptions here: https://www.cs.ucr.edu/~eamonn/time_series_data/.  Some datasets seem to have a small degree of possible coordinate sensitivity, such as the spectrogram data, while other would vary significantly with scale, placement of the signal within a collection window, and image orientation.  (This is also true of MNIST).  Please review the table below, and help me understand the sensitivity of your method to data collection errors.  This gets at the issue of generalizability / fragility of the method.  In the table, the first column has the dataset name, the second column records whether or not your method was the top score in your reported results, the third records a (possibly incorrect) sense of how sensitive the data collection is to getting all the coordinates right, and the fourth is an explanation of the assessment.

| Dataset | NeurRL | Coordinate Sensitivity | Rationale for sensitivity assessment |
|---------|-------------|----------------------|-------------|
| Coffee | Top | Low | Based on spectrographic measurements which are inherently coordinate-independent. The frequency patterns would be consistent regardless of measurement setup, as long as calibration is maintained. |
| ECG | -- | Medium | While heart rhythms have characteristic shapes, the absolute voltage measurements depend on electrode placement. However, the relative patterns are fairly robust across different setups. |
| Gun Point | -- | High | Heavily dependent on camera angle, distance calibration, and the defined origin point for measuring hand position. The distinction between gun/no-gun relies on precise spatial coordinates. |
| ItalyPow.Dem | Top | Low | Power demand measurements are absolute values independent of coordinate systems. The readings represent actual power consumption regardless of measurement setup. |
| Lightning2 | -- | Low (?) | While electromagnetic measurements depend on sensor placement, the characteristic patterns of lightning types are relatively robust across different setups. |
| CBF | -- | Low | As a synthetic dataset designed to test pattern recognition, the shapes (Cylinder-Bell-Funnel) are meaningful in their relative values rather than absolute coordinates. |
| Face Four | Top | High | Face profile traces are highly dependent on camera angle, distance, and orientation. Changes in perspective would significantly alter the time series. |
| Lightning7 | Top | Low (?) | Similar to Lightning2, with moderate dependence on sensor placement but relatively robust patterns. |
| OSU Leaf | Top | High | Leaf contour measurements depend heavily on orientation, scale, and starting point of the trace. Different coordinate systems would produce very different time series. |
| Trace | -- | Low | As synthetic nuclear instrument data, the patterns represent relative changes that are meaningful independent of absolute coordinate systems. |
| WordsSyn | Top | High | Pen trajectories are highly dependent on writing orientation, scale, and starting position. The coordinate system directly affects the recorded patterns. |
| OliveOil | Top | Low | Spectroscopic measurements are independent of coordinate systems. The chemical signatures would be consistent across different spectrometers (after calibration). |
| StarLightCurves | Top | Low | Brightness measurements are relative values that remain meaningful regardless of the telescope's exact positioning (assuming proper astronomical calibration). |

Question 2:  Can you explain "leakage" more clearly?  The coordinate dependence an example above creates an implicit bias in the data, for instance limited field-of-view and imaging system orientation in MNIST and handwriting digitization; is this an example of "label leakage"?

---

> ### Author Response · Authors · 2024-11-22
> **Responses to the Weakness**
>
> Thank you for your thoughtful review. We are so grateful that you find our work as exciting as we do. We would like to address the points brought up below.
>
> >Weaknesses:
> >The approach may require significant data invariance (orientation of images, sampling regularity). Many important datasets have these properties, although image data often does not. See questions for further clarification.
> >As a demonstration of the advantage of this approach as an interpretable and explainable method, it would be helpful to have the interpretation of rules on each dataset from UCR discussed in more detail. How interpretable are the rules in terms of the data patterns? Are there particular datasets or data types for which rules emerge that provide insight or match intuition?
>
> Thank you for the comments. In the learned rules, we use patterns as atoms to explain the ground truth class. We explained the semantics of rules in lines 399 to 403, lines 475 to 480, and lines 507 to 510 for synthetic data, time series data, and MNIST data, respectively. For UCR datasets, we also explain rules with time information. The conjunction of patterns should be very discriminative to describe the ground truth class from all data. Besides, the shape of the patterns is not limited by a specific datatype or dataset. Indeed, we can choose the data with invariance to induce rule and explain the data. We can explore real-world healthcare data to apply NeurRL for rule generation. In the time series domain, data types such as ECG signals in cardiology and glucose level trends in laboratory results could be utilized to apply AI in healthcare. In the image domain, medical imaging data such as X-rays could also be leveraged.

---

> ### Author Response · Authors · 2024-11-22
> **Responses to the Questions**
>
> >Questions:
> >Please clarify the generalizability of your approach. Do you expect the rule learning method to be sensitive to data errors that arise naturally in real-world collection, such as uncalibrated sensors or aspect/focus variation in images, or frame rate in video?
> >Consider the following comment, if that helps clarify the objective of the request.
> > ...
> >Dataset	NeurRL	Coordinate Sensitivity	Rationale for sensitivity assessment
> >...
> >Lightning2	--	Low (?)	While electromagnetic measurements depend on sensor placement, the characteristic patterns of lightning types are relatively robust across different setups.
> >CBF	--	Low	As a synthetic dataset designed to test pattern recognition, the shapes (Cylinder-Bell-Funnel) are meaningful in their relative values rather than absolute coordinates.
> >Face Four	Top	High	Face profile traces are highly dependent on camera angle, distance, and orientation. Changes in perspective would significantly alter the time series.
> >Lightning7	Top	Low (?)	Similar to Lightning2, with moderate dependence on sensor placement but relatively robust patterns.
> >OSU Leaf	Top	High	Leaf contour measurements depend heavily on orientation, scale, and starting point of the trace. Different coordinate systems would produce very different time series.
> >Trace	--	Low	As synthetic nuclear instrument data, the patterns represent relative changes that are meaningful independent of absolute coordinate systems.
> >WordsSyn	Top	High	Pen trajectories are highly dependent on writing orientation, scale, and starting position. The coordinate system directly affects the recorded patterns.
> >OliveOil	Top	Low	Spectroscopic measurements are independent of coordinate systems. The chemical signatures would be consistent across different spectrometers (after calibration).
> >StarLightCurves	Top	Low	Brightness measurements are relative values that remain meaningful regardless of the telescope's exact positioning (assuming proper astronomical calibration).
>
> We greatly appreciate your supported data analysis based on the viewpoint of the Coordinate Sensitivity. We added the correct link to the UCR archive in the paper. Our model is based on neural networks; therefore, some errors or variations in different subsequences with similar shapes may still cause them to be regarded as the same patterns. For example, please see Figure 3 (a), the patterns in the red color are not the same but have similar trends and mean values.
>
> In NeurRL, we use min-max normalization to process time series data before running NeruRL. After the min-max-normalization, the instances in different measurement scales would be the same scale. Besides min-max normalization, we can also consider Z-normalization [1], which ensures that all elements of the input vector are transformed into the output vector whose mean is approximately 0 while the standard deviation is in a range close to 1.
>
> Besides the normalization, NeurRL learns rules for describing the ground truth class instances in the data-driven model. If multiple time series instances share the same class but differ in coordinate scales, it is beneficial to include multiple instances from both positive and negative classes within the same coordinate scale. This allows the model to learn all the discriminative features with different coordinate scales effectively, even when instances exist under varying coordinate scales.
>
> Reference:
>
> [1] Dina Q. Goldin, Paris C. Kanellakis: On Similarity Queries for Time-Series Data: Constraint Specification and Implementation. CP 1995: 137-153.
>
> >Question 2: Can you explain "leakage" more clearly? The coordinate dependence an example above creates an implicit bias in the data, for instance limited field-of-view and imaging system orientation in MNIST and handwriting digitization; is this an example of "label leakage"?
>
> Topan et al. [1] defined label leakage as The inability to map raw inputs to symbolic variables without explicit supervision. In our paper, we describe label leakage as follows: Creating rules from raw data needs a perception model to locate the object and label the object as a symbolic atom before the rule-learning process. Avoiding label leakage indicates that the pattern labels should not be provided for learning rules to describe the ground truth class. For example, we do not need to label all possible significant patterns, such as a growth in a specific region of a time series, to induce the rules to describe the ground truth class. Hence, NeurRL is very suitable for learning explainable knowledge in raw input data because we cannot enumerate all possible patterns in a raw input. Please let us know if this answers your question. We would be happy to discuss further.
>
> Reference:
>
> [1] Sever Topan, David Rolnick, Xujie Si: Techniques for Symbol Grounding with SATNet. NeurIPS 2021: 20733-20744

---

> > ### Comment · Reviewer_6MqZ · 2024-12-01
> >
> > Thank you for your thoughtful response.  I will maintain my assessment that this is a good paper, and should be accepted.

---

> ### Author Response · Authors · 2024-12-02
>
> We sincerely thank you once again for your valuable support and insightful suggestions!

---

### Meta-Review · Area_Chair_z9Re · 2024-12-20

**Metareview:**

The paper makes an interesting contribution: dropping the need for pre-trained neural networks within neuro-symbolic learning approaches. To this extent, it combines VAE and differentiable k-means into a differentiable rule learning approach. The reviewers agree that this problem is relevant, though they also point out some downsides. For instance, one reviewer points out that some related work is missing and needs to be discussed. It is also argued that the paper should be polished so that it is clear (1) the present paper does not touch the symbol grounding problem and (2) the impression has to be avoided, the present paper is the first one to learn a neural network together with a rule set en3ent, which is definitely not true. However, what seems to be true the present paper pushes this line of work towards fully differentiable rule learning. This is interesting and important, and overall, I agree with the reviews that this warrants publication. The combination of the submodules together with a fully differentiable rule learning approach seems to be novel.

**Additional Comments On Reviewer Discussion:**

The discussion arose from problems and questions that had been raised in the reviews. They actually helped to clarify some of the concerns raised. It also led to increased scores.

---

### Decision · Program_Chairs · 2025-01-22

Accept (Poster)